# BMAL1 and ARNT enable circadian HIF2α responses in clear cell renal cell carcinoma

Rebecca M. Mello [1,2], Diego Gomez Ceballos [1,2], Colby R. Sandate [3], Sicong Wang[1,2], Celine Jouffe [4,5], Daniel Agudelo[4], Nina Henriette Uhlenhaut[4,6], Nicolas H. Thomä [3], M. Celeste Simon [7,8] & Katja A. Lamia [1,2]✉

Circadian disruption enhances cancer risk, and many tumors exhibit disordered circadian gene expression. We show rhythmic gene expression is unexpectedly robust in clear cell renal cell carcinoma (ccRCC). The core circadian transcription factor BMAL1 is closely related to ARNT, and we show that BMAL1-HIF2α regulates a subset of HIF2α target genes in ccRCC cells. Depletion of *BMAL1* selectively reduces HIF2α chromatin association and target gene expression and reduces ccRCC growth in culture and in xenografts. Analysis of pre-existing data reveals higher *BMAL1* in patient-derived xenografts that are sensitive to growth suppression by a HIF2α antagonist (PT2399). BMAL1-HIF2α is more sensitive than ARNT-HIF2α is to suppression by PT2399, and the effectiveness of PT2399 for suppressing xenograft tumor growth in vivo depends on the time of day at which it is delivered. Together, these findings indicate that an alternate HIF2α heterodimer containing the circadian partner BMAL1 influences HIF2α activity, growth, and sensitivity to HIF2α antagonist drugs in ccRCC cells.

Many tumors exhibit disruption of circadian rhythms[1], and deletion of the clock component BMAL1 exacerbates tumor burden in several genetically engineered mouse models of cancer[2,3]. However, circadian disruption is not universally observed in cancer cells, and BMAL1 depletion improves outcomes in some cancer models[4]. It has been unclear why genetic deletion of BMAL1 enhances the growth of some tumors and suppresses others.

The Von Hippel-Lindau (VHL) ubiquitin ligase is inactivated in 50–85% of clear cell renal cell carcinomas (ccRCC)[5–8]. VHL targets hypoxia inducible factors 1 alpha (HIF1α) and 2 alpha (HIF2α, a.k.a. EPAS1) for degradation[9]. HIF1α and HIF2α are basic helix-loop-helix and PER-ARNT-SIM domain (bHLH-PAS) transcription factors that bind DNA with a common heterodimer partner HIF1β (a.k.a. ARNT), and increase the expression of genes involved in metabolism, proliferation, and angiogenesis[7,8,10–12]. Suppression of HIF2α is required for VHL to

inhibit ccRCC tumor growth[13,14], highlighting the oncogenic role of HIF2α in ccRCC.

In mammals, circadian clocks comprise a transcription-translation feedback loop, centered around the heterodimeric transcription factor complex containing CLOCK and BMAL1[15]. CLOCK and BMAL1 are bHLH-PAS transcription factors and are closely related to ARNT and HIF2α[16,17] (Fig. 1A, B). At the time of its initial characterization, BMAL1 was found to be dispensable for developmental processes in which HIFs are key players, and was therefore considered not to be a relevant partner for HIF alpha subunits[18–21]. This impression was reinforced when X-ray crystal structures described divergent arrangements of the bHLH and PAS domain interfaces for CLOCK-BMAL1 and for HIFα-ARNT complexes[16]. However, the arrangement of BMAL1 PAS domains is flexible even within CLOCK-BMAL1 heterodimers[22], and BMAL1 can activate transcription via

[1]Department of Molecular and Cellular Biology, Scripps Research Institute, La Jolla, CA, USA. [2]Department of Molecular Medicine, Scripps Research Institute, La Jolla, CA, USA. [3]Friedrich Miescher Institute for Biomedical Research, Basel, Switzerland. [4]Institute for Diabetes and Endocrinology (IDE), Helmholtz Munich, Neuherberg, Germany. [5]Institute for Diabetes and Cancer (IDC), Helmholtz Munich, Neuherberg, Germany. [6]Metabolic Programming, TUM School of Life Sciences & ZIEL Institute for Food and Health, Freising, Germany. [7]Perelman School of Medicine, Abramson Family Cancer Research Institute, Philadelphia, PA, USA. [8]Department of Cell and Developmental Biology, University of Pennsylvania, Philadelphia, PA, USA. ✉e-mail: klamia@scripps.edu

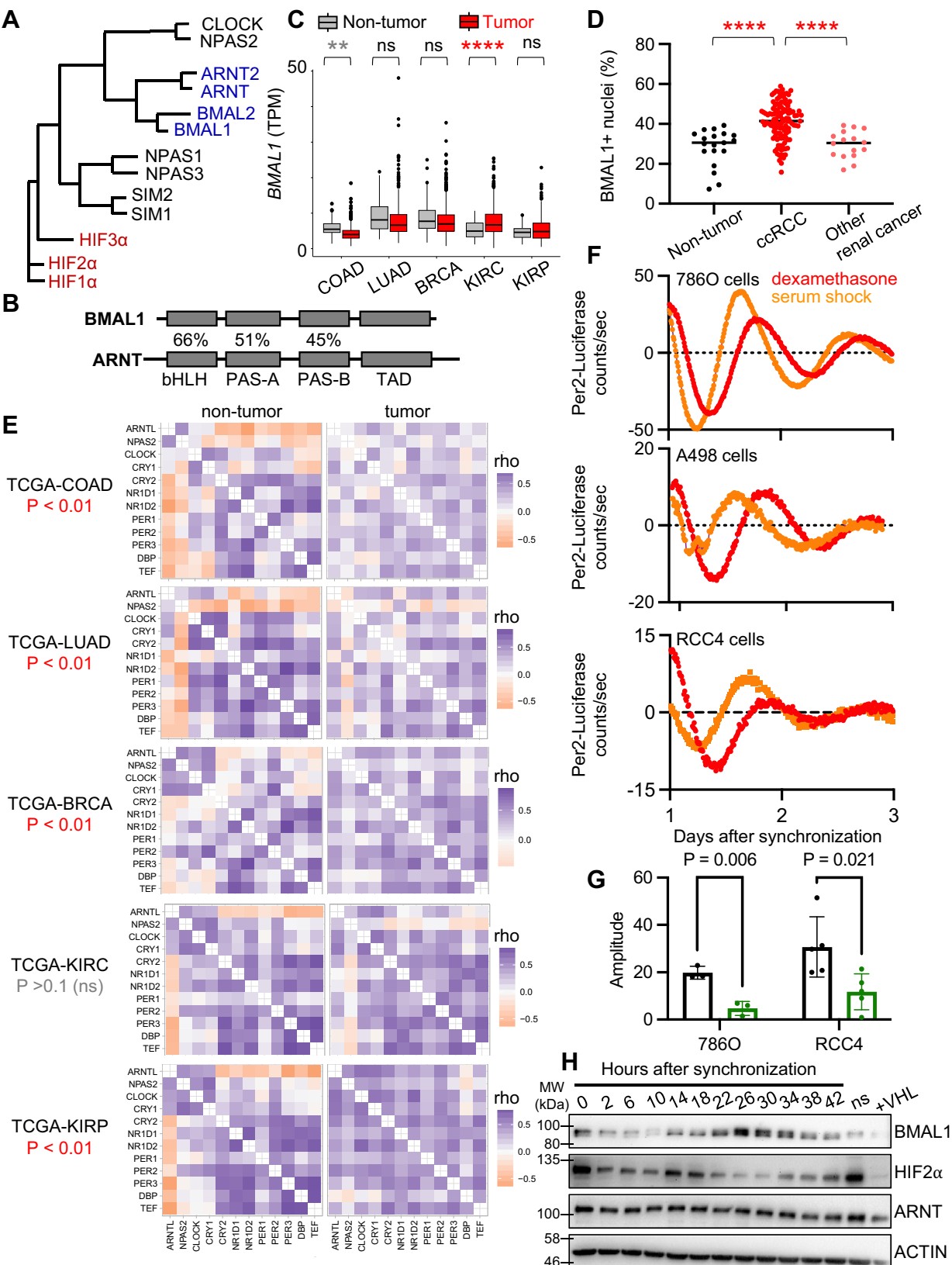

**Figure A** — phylogenetic tree: CLOCK, NPAS2, ARNT2, ARNT, BMAL2, BMAL1, NPAS1, NPAS3, SIM2, SIM1, HIF3α, HIF2α, HIF1α

**Figure B** — BMAL1 and ARNT domain comparison: bHLH 66%, PAS-A 51%, PAS-B 45%, TAD

**Figure C** — *BMAL1* (TPM); Non-tumor (gray), Tumor (red); COAD **, LUAD ns, BRCA ns, KIRC ****, KIRP ns

**Figure D** — BMAL1+ nuclei (%); Non-tumor, ccRCC, Other renal cancer; **** , ****

**Figure E** — non-tumor vs tumor correlation heatmaps (rho); TCGA-COAD P < 0.01, TCGA-LUAD P < 0.01, TCGA-BRCA P < 0.01, TCGA-KIRC P > 0.1 (ns), TCGA-KIRP P < 0.01

**Figure F** — Per2-Luciferase counts/sec; 786O cells, A498 cells, RCC4 cells; dexamethasone (red), serum shock (orange); Days after synchronization

**Figure G** — Amplitude; 786O P = 0.006, RCC4 P = 0.021

**Figure H** — Hours after synchronization; BMAL1, HIF2α, ARNT, ACTIN

hypoxia response elements (HREs) in cooperation with HIF alpha subunits[23,24]. Accumulating evidence indicates that BMAL1 is an important partner for HIF1α-dependent hypoxic responses[23–25]. Together, these findings motivate a reconsideration of the possible physiological relevance of a more diverse set of bHLH-PAS heterodimer pairings.

Small molecules that interact with a pocket in the PAS-B domain of HIF2α disrupt the formation of HIF2α heterodimers and are used to treat ccRCC. Variability in responses to these drugs can be caused by mutations surrounding their binding site in HIF2α or ARNT in some cases but is not generally understood[14,26–28]. Here, we demonstrate that BMAL1 forms a transcriptionally active heterodimer with HIF2α in

**Fig. 1 | BMAL1 is elevated and active in ccRCC. A** Phylogenetic tree for bHLH-PAS proteins. **B** Percent sequence identity for bHLH and PAS domains in BMAL1 and ARNT. **C** Detection of *BMAL1* (transcripts per million, TPM) in RNA sequencing data from tumors and adjacent normal tissues in cancer genome atlas projects: color-ectal adenocarcinoma (COAD, n = 82 non-tumor, n = 962 tumor), lung adeno-carcinoma (LUAD, n = 59 non-tumor, n = 539 tumor), breast cancer (BRCA, n = 113 non-tumor, n = 1111 tumor), kidney clear cell renal cell carcinoma (KIRC, n = 72 non-tumor, n = 541 tumor), and renal papillary carcinoma (KIRP, n = 100 non-tumor, n = 872 tumor). **D** Nuclear BMAL1 protein detected in human kidney biopsy samples from ccRCC (n = 138), other renal cancers (n = 42), and non-tumor kidney tissue (n = 30). **E** Clock correlation distance (CCD) heatmaps calculated from RNA sequencing data from tumors and adjacent normal tissues in the Cancer Genome Atlas projects. In (**E**) one-sided p-values for non-tumor vs. tumor samples are cal-culated from permutation testing as described in detail in ref. 1. **F** Luminescence detected in cells expressing destabilized luciferase under the control of the PER2 promoter in 786O, A498, or RCC4 cell lines in which circadian rhythms were synchronized by treatment with dexamethasone (red) or horse serum (orange). **G** Quantitation of the rhythmic amplitude for data as in (**F**) for 786 O (n = 3 biolo-gical replicates per condition) or RCC4 (n = 4 biological replicates per condition) cells expressing *Per2-Luciferase* with (green) or without (black) concomitant expression of VHL and synchronized with dexamethasone. Error bars represent s.d. P values calculated by two-sided t-tests. **H** Detection of the indicated proteins by immunoblot in cell lysates prepared from 786O cells at the indicated times after treatment with dexamethasone or without synchronization (ns) or expressing VHL. Data represent one of two independent experiments with similar results. The samples derive from the same experiment but one gel for BMAL1, HIF2α, and ACTIN, and another for ARNT and ACTIN were processed in parallel. In (**C**), box-plots depict the median and interquartile range (IQR), whiskers extend either to the minimum or maximum data point or 1.5 × IQR beyond the box, whichever is shorter. Outliers (values beyond the whisker) are shown as dots. In (**C,D**) **P = 0.00149, ****P < 0.0001 by two-way ANOVA. Source data are provided as a Source Data file.

ccRCC-derived cells and contributes to HIF2α-driven gene expression, cell and tumor growth. Furthermore, we show that BMAL1-HIF2α het-erodimers are more sensitive to disruption by the HIF2α antagonist PT2399 than ARNT-HIF2α heterodimers are. Finally, we demonstrate that the suppression of xenograft tumor growth by PT2399 depends on the time of day at which it is delivered. Together, these findings indicate that BMAL1-HIF2α heterodimers enable circadian regulation of HIF2α activity and responses to HIF2α antagonist drugs in ccRCC.

## Results

### ccRCC tumors maintain robust circadian rhythms

Using data from the Clinical Proteomic Tumor Analysis Consortium (CPTAC) and the Cancer Genome Atlas (TCGA), we find that *BMAL1* expression is higher in samples collected from ccRCC tumor biopsies than it is in non-tumor kidney tissue, whereas *BMAL1* expression in several other tumor types is either reduced or unchanged from normal samples of the same tissue type (Fig. 1C and Supplementary Fig. S1). Increased *BMAL1* in ccRCC compared to non-tumor biopsies remains statistically significant when only adjacent samples from the same patients are included in the analysis, suggesting that elevated *BMAL1* in ccRCC samples is not an artifact of tissue collection time or differential sample processing (Supplementary Fig. S1B). *ARNT2* expression is reduced in ccRCC; *ARNT* and *BMAL2* are not significantly different in ccRCC compared to adjacent kidney biopsies from the same patients (Fig S1B). Furthermore, we find that BMAL1 protein is present in a greater proportion of nuclei, and therefore likely to be more active, in ccRCC tumors compared to non-tumor kidney samples and compared to other types of renal cancer (Figs. 1D and S1C).

Correlated expression of twelve genes that are strongly driven by circadian rhythms has been established as a readout of circadian robustness, and this has been used to demonstrate that many tumors have disrupted rhythms[1]. Using this measure, we find that circadian rhythmicity is not disrupted in ccRCC, in contrast to other tumor types examined, including papillary RCC, a distinct form of renal cancer that is not driven by HIF2α (Figs. 1E and S2A).

To monitor circadian rhythmicity in ccRCC cell lines in cell cul-ture, we used destabilized luciferase reporters that have been widely studied[29]. We treated ccRCC cell lines with either 50% horse serum or with 1 μM dexamethasone to synchronize their circadian clocks and observed robust rhythms in luciferase activity driven by a *Per2-Luciferase* reporter in 786O, A498, and RCC4 cells (Fig. 1F). Further, we found that the amplitude of circadian rhythmicity is reduced by re-introduction of wildtype VHL into 786 O or RCC4 cells (Fig. 1G and Supplementary Fig. S2B), suggesting that enhanced HIF2α expression may contribute to the unexpectedly robust circadian rhythms observed in ccRCC. To further characterize circadian rhythms in ccRCC cells and their potential influence on HIF, we measured the levels of endogenous BMAL1, ARNT, and HIF2α in protein lysates prepared from synchronized 786 O cells over two consecutive circa-dian cycles. We measured rhythmic accumulation of BMAL1 protein as expected and also observed rhythmic accumulation of HIF2α protein. Notably, the peaks of BMAL1 and HIF2α protein accumulation are approximately antiphase (i.e., separated by ~12 h, Fig. 1H).

### Depletion of *BMAL1* suppresses RCC cell growth in vitro and in vivo

Data from the Cancer Dependency Map (DepMap)[30,31] shows that deletion of *BMAL1* reduces survival of RCC cells (Fig. 2A), indicating that BMAL1 supports growth and/or survival of cells in this lineage. Some RCC-derived cell lines (e.g., 786 O) are highly dependent on ARNT, while others (e.g., 769P, RCC10RGB) are less so, such that on average RCC-derived cell lines are not significantly more dependent on ARNT than on the control gene HPRT1 (Fig. 2A, B). Genes that act in concert to support cell growth often exhibit correlated dependencies[32]. In RCC-derived cell lines, dependencies for *ARNT* and *EPAS1* are strongly correlated as expected based on their well-established heterodimeric activation of HIF2α-dependent gene expression. Notably, dependencies for *BMAL1* and *EPAS1* are also strongly correlated in RCC cell lines, while no such correlations are detected for the dependencies of *ARNT2* or *BMAL2* with *EPAS1* dependency across RCC cell lines (Fig. 2B and Supplementary Fig. S3). Together, these data suggest that both BMAL1 and ARNT cooperate with HIF2α to support the growth and survival of RCC cells.

To directly investigate whether BMAL1 promotes the growth and survival of ccRCC cells, we measured clonogenicity in three cell lines derived from ccRCC tumor biopsies (786O, RCC4, and A498) in which we used shRNA to deplete the expression of endogenous BMAL1. In all three cell lines, depletion of *BMAL1* reduces colony formation in cells plated at low density (Fig. 2C–E). To evaluate the impact of BMAL1 in vivo, we generated xenograft tumors from two ccRCC cell lines in immunocompromised murine hosts. Depletion of *BMAL1* suppressed the growth of tumors derived from 786O or A498 cells in vivo (Fig. 2F). Clinically, ccRCC is much more common in men than in women[5]. Here, we observed no difference in the growth of cell-derived xenograft tumors implanted in male or female hosts (Fig. 2F), suggesting that factors contributing to sexual dimorphism in ccRCC are not present in this model system.

### BMAL1 forms a transcriptionally active heterodimer with HIF2α

Given its homology with ARNT, the most straightforward mechanistic hypothesis for the cooperation of BMAL1 with HIF2α to support ccRCC cell growth would be as a partner in hetero-dimeric transcriptional activation. We and others have shown that BMAL1 can interact with HIF1α[23-25,33]. To determine whether BMAL1 can form a stable complex with HIF2α in vitro, we co-expressed and purified the two proteins from insect cells. BMAL1

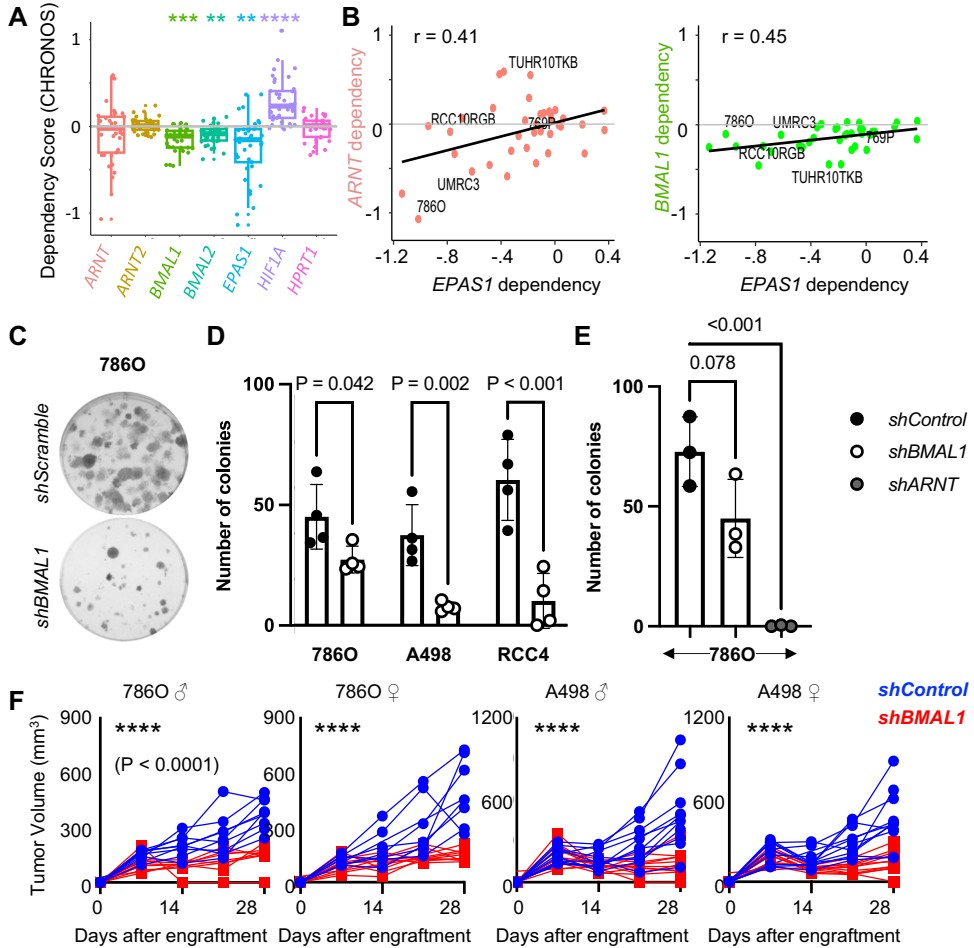

**Fig. 2 | BMAL1 promotes growth and survival in ccRCC cells.** Dependency (CHRONOS) scores (**A**) and correlations thereof (**B**) for bHLH-PAS members in n = 37 RCC cell lines from DepMap[30,31]. Boxplots depict the median and interquartile range (IQR), whiskers extend either to the minimum or maximum data point or 1.5 × IQR beyond the box, whichever is shorter. In (**A**), p values calculated from multiple paired t tests with *HPRT1* as the reference group, adjusted for multiple comparisons using the Holm method, were: *BMAL1*: 0.001, *BMAL2*: 0.007, *EPAS1*: 0.006, *HIF1A*: $1.47e^{-6}$. Representative images (**C**) and quantification (**D,E**) of colonies stained with crystal violet 10–16 days after plating 250 cells expressing the indicated plasmids per well. Data represent the mean ± s.d. for four (**D**) or three (**E**) independent experiments, each of which used three wells per condition. *P < 0.05,

**P < 0.01, ***P < 0.001 by two-way ANOVA with Tukey's correction for multiple hypothesis testing. **F** Volume of xenograft tumors grown in flanks of male or female NIH-III Nude mice from implanted 786O or A498 cells expressing indicated shRNAs. n = 10 xenografts were initiated for each group; up to two xenografts that failed to establish were excluded from analysis for each group. Weekly measurements of individual tumor volumes are shown. ****P < 0.0001 for *shBMAL1* vs *shControl* by linear regression. Details of linear regression statistics: 786 O male: F = 30.13, DFn = 1, DFd = 75; 786O female: F = 41.58, DFn = 1, DFd = 78; A498 male: F = 26.01, DFn = 1, DFd = 96; A498 female: F = 27.02, DFn = 1, DFd = 96. Source data are provided as a Source Data file.

and HIF2α co-eluted during heparin chromatography, and SDS-PAGE analysis indicated that they formed a stoichiometric complex (Fig. 3A). Analysis by mass photometry of the purified sample (Fig. 3B) further confirmed that the two proteins formed a stable heterodimeric complex, even at low concentration (20 nM). To evaluate the potential for BMAL1 to partner with HIF2α in ccRCC cells, we expressed FLAG-tagged bHLH-PAS family members ARNT, ARNT2, BMAL1, or BMAL2 in 786O cells, in which endogenous wild-type HIF2α is stable. By immunoprecipitation of the FLAG tag, we found that BMAL1 interacts with endogenous HIF2α in addition to endogenous CLOCK, its canonical partner (Fig. 3C). In contrast, FLAG-ARNT interacts with HIF2α but does not bind CLOCK, consistent with prior reports[23]. To evaluate whether BMAL1 can cooperate with HIF2α to activate target gene expression, we used a luciferase reporter under the control of a hypoxia response element derived from the *PGK1* promoter region (*HRE-Luciferase*). We demonstrate that overexpression of either ARNT or BMAL1 enhances activation of *HRE-Luciferase* by HIF2α (Fig. 3D) in kidney cells.

## BMAL1 regulates HIF2α target gene expression in ccRCC cells

To measure the contributions of endogenous ARNT and BMAL1 to HIF2α-driven gene expression in ccRCC cells, we sequenced RNA prepared from 786O cells in which either ARNT or BMAL1 was depleted by shRNA. Efficient depletion of BMAL1 or ARNT was confirmed by Western blot and did not reduce HIF2α protein in 786 O cells (Fig. 4A). A slight increase in ARNT protein observed upon depletion of BMAL1 and vice versa may indicate compensatory upregulation of the remaining homolog. We used DESeq2[34] to identify transcripts that were significantly altered and found a striking overlap between the genes affected by loss of ARNT and those affected by loss of BMAL1 (Fig. 4B). Because HIF2α and BMAL1 are expected to primarily activate the expression of their transcriptional targets, we focused on genes that exhibit significantly decreased expression upon depletion of ARNT or BMAL1 as more likely direct targets (Fig. 4C): 42.7% or 54.3% of the transcripts that were significantly decreased by *shBMAL1* or by *shARNT* were decreased by both shRNAs (Fig. 4C, D). Hallmark gene sets generated from multiple primary experiments represent the transcripts regulated by pathways of interest with high confidence

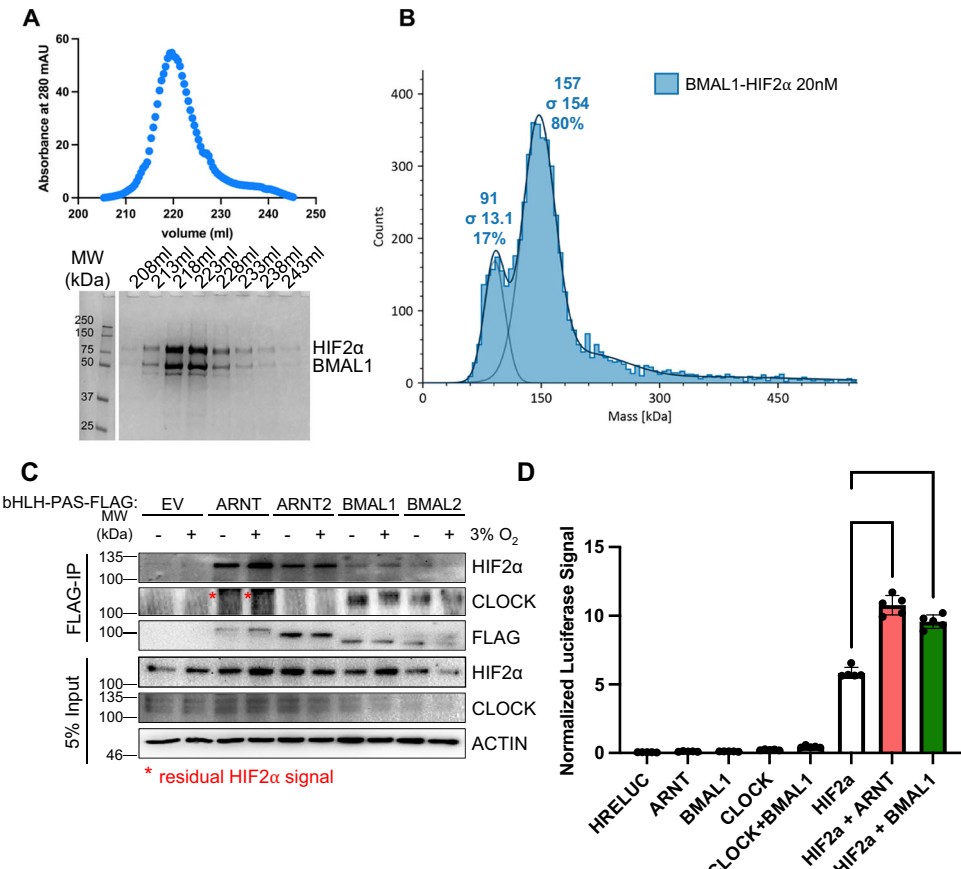

**Fig. 3 | BMAL1 and HIF2α form an active heterodimer. A** Heparin chromatography elution of BMAL1 and HIF2α co-expressed in insect cells. SDS-PAGE analysis shows a co-eluted stoichiometric complex of BMAL1-HIF2α from one of two independent experiments with similar results. **B** Mass photometry of purified BMAL1-HIF2α complex. A minor peak centered at 91 kDa corresponds to the molecular weight of HIF2α, suggesting that it is in slight excess. The major peak, centered at 157 kDa, is consistent with the calculated molecular weight for the BMAL1-HIF2α heterodimer. **C** Detection of endogenous HIF2α and CLOCK and of ectopically expressed FLAG-tagged class I bHLH-PAS proteins by immunoblot of lysates (input)

or complexes purified with an anti-FLAG antibody from 786O cells expressing the indicated plasmids. The data represent one of three independent experiments with similar results. **D** Relative luminescence units detected in HEK293T cells expressing luciferase under the control of a hypoxia-responsive element with the indicated additional plasmids. ****P < 0.0001 by one-way ANOVA with Tukey's correction for multiple comparisons. Data in (**D**) depict the mean ± s.d. for n = 5 biological replicates from one of three independent experiments with similar results. Source data are provided as a Source Data file.

across experimental conditions[35]. We examined the expression of 200 transcripts in the Hallmark HYPOXIA gene set[35] using gene set enrichment analysis (GSEA)[36] and found that they are robustly impacted by depletion of either *ARNT* or *BMAL1* (Fig. 4E–G). In a separate experiment, we used 786 O cells expressing wildtype VHL (WT8 cells) to highlight transcripts impacted by VHL-dependent suppression of HIF2α. Notably, all transcripts altered by depletion of *BMAL1* in 786 O cells were also affected by rescue of VHL (Fig. 4H and Supplementary Fig. S4).

We took advantage of data from a previous study[26] that examined the impact of the HIF2α antagonist drug PT2399 on gene expression in patient-derived xenograft (PDX) tumors to ask how transcripts that are specifically dependent on either ARNT or BMAL1 are affected by disruption of HIF2α heterodimers in vivo. We find that genes that are reduced by depletion of ARNT and/or BMAL1 in 786O cells exhibit significantly lower expression in PDX samples that are sensitive to growth inhibition by PT2399 when treated with the drug compared to those treated with vehicle alone (Fig. 4I–L). A more detailed analysis reveals that ARNT-specific targets are enriched in genes related to hypoxia response, ribosome, and metabolism pathways and have higher GC content, while BMAL1-specific targets are enriched in genes related to mitosis, intracellular transport, proteasome, and circadian rhythm pathways and have greater

transcript length and longer 5' untranslated regions (Fig. 4M, N and Supplementary Fig. S5). Similar outcomes were observed upon depleting ARNT or BMAL1 in A498 cells (Supplementary Figs. S6 and S7). Together, these findings show that endogenous ARNT and BMAL1 regulate the expression of overlapping and distinct HIF2α target genes in ccRCC patient-derived cells.

### BMAL1 influences HIF2α recruitment to chromatin
HIF2α promotes transcription as part of a heterodimeric complex that interacts with hypoxia response elements (HREs: 5'-N(G/A)CGTG-3'), which are closely related to the canonical E-box sequence bound by BMAL1-CLOCK heterodimers (5'-CACGTG-3'). This suggests that BMAL1 could influence HIF2α target gene expression through diverse mechanisms, including transcriptional activation by BMAL1-HIF2α heterodimers and competition for sites that match both recognition sequences. Subsets of target sites are likely preferentially regulated by alternate bHLH-PAS heterodimers with distinct sequence preferences. To characterize the localizations of endogenous BMAL1 and HIF2α in native chromatin and how these are impacted by depletion of BMAL1, we sequenced genomic DNA associated with BMAL1 or HIF2α in 786O cells. We used MACS2[37] to identify 1813 and 1204 genomic regions enriched in chromatin purified with BMAL1 or HIF2α, respectively (Fig. 5A, B). Consistent with prior reports[38,39], genomic regions

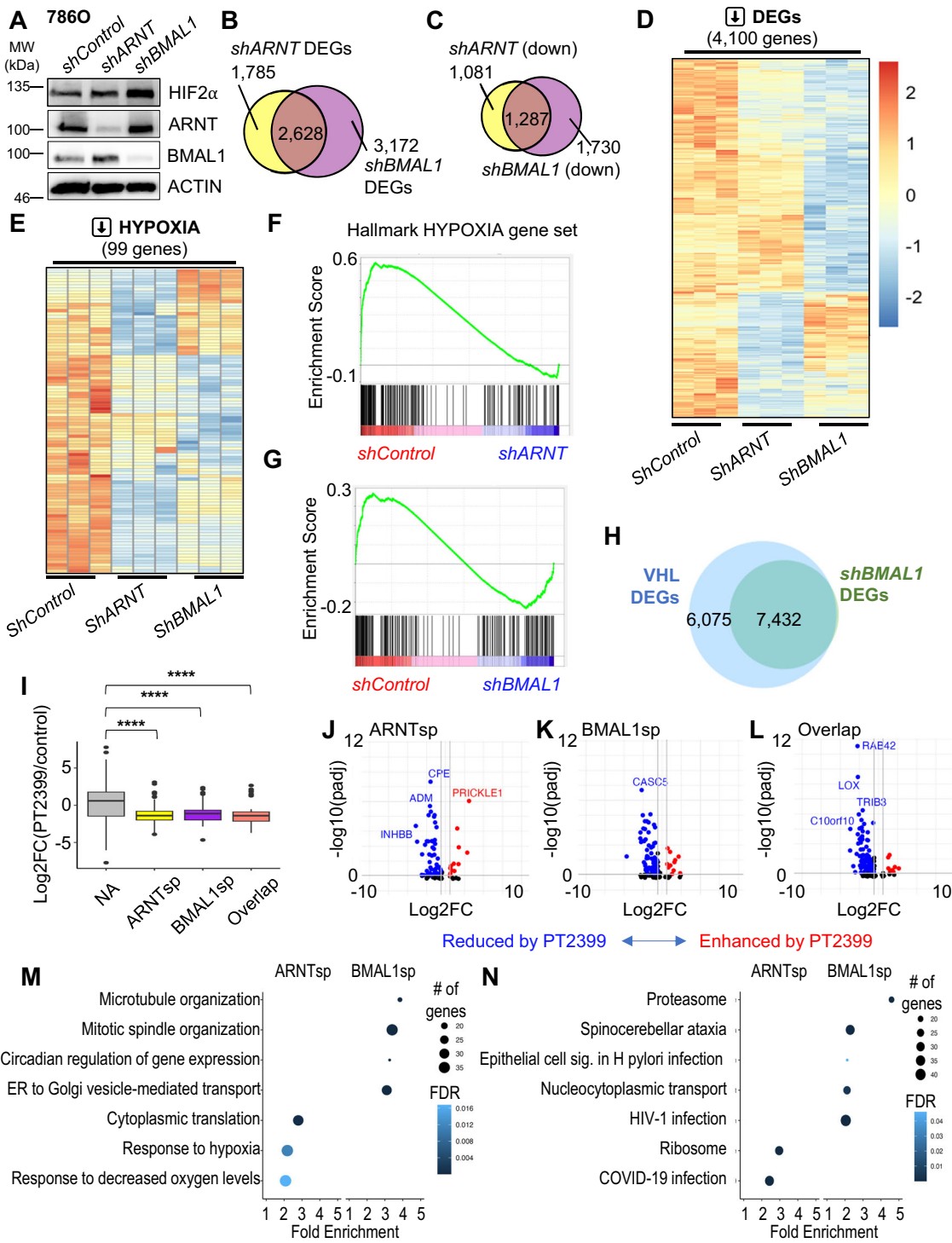

associated with BMAL1 and HIF2α are enriched in promoters and introns (Supplementary Fig. S8A). 336 loci were identified as co-occupied by BMAL1 and HIF2α, representing 18.5% or 27.9% of the sites associated with BMAL1 or HIF2α, respectively (Fig. 5A). We used Hypergeometric Optimization of Motif EnRichment (HOMER)[40] to identify sequence motifs that are enriched in chromatin associated with HIF2α and/or BMAL1. The top motifs identified include those that have been defined biochemically to be preferentially associated with bHLH-PAS transcription factors, including CLOCK, BMAL1, ARNT (a.k.a. HIF-1b), HIF1α, and HIF2α (Fig. 5C). Notably, motifs associated with CLOCK and NPAS were enriched uniquely in the BMAL1 cistrome. These findings provide confidence in the sensitivity and specificity of

the data and demonstrate that BMAL1 and HIF2α co-occupy a sizeable fraction of each of their cistromes in 786O ccRCC cells.

Depletion of BMAL1 reduced chromatin association of both BMAL1 and HIF2α at many sites that were occupied in control cells and reduced the number of significantly enriched loci detected in chromatin purified with BMAL1 (Figs. 5B, D and Supplementary Fig. S8). Of the 1813 BMAL1-associated peaks detected in *shControl*-expressing cells, 702 (38.7%) were detected in cells expressing *shBMAL1*. The intensity of the peaks that remained was reduced, suggesting that the remaining signal is coming from residual BMAL1 protein expression in *shBMAL1*-expressing cells. Of those 702 peaks, 98 were identified as co-occupied by BMAL1 and HIF2α, representing 12.7% or 9.0% of the sites

**Fig. 4 | Endogenous BMAL1 contributes to HIF2α target gene expression in RCC cells. A** Detection of HIF2α, ARNT, BMAL1, and ACTIN by immunoblot in 786O cells expressing the indicated shRNAs. The samples derive from the same experiment but one gel for BMAL1, HIF2α, and ACTIN, and another for ARNT and ACTIN were processed in parallel. Data from one of two independent experiments with similar results. Venn diagrams (**B, C**) and heatmaps (**D, E**) depicting all differentially expressed genes (DEGs) (**B**, significantly downregulated genes (**C, D**) or downregulated genes in the Hallmark HYPOXIA gene set (**E**) in 786O cells expressing the indicated shRNAs (n = 3 distinct samples for each condition). DEGs were identified using DESeq2 with a false discovery rate (FDR) cutoff of 0.1. Enrichment plots showing the impact of *shARNT* (**F**) or *shBMAL1* (**G**) on genes in the Hallmark HYPOXIA gene set. **H** Venn diagram depicting overlap of DEGs in 786O cells expressing VHL (WT8 cells) or expressing *shBMAL1*. **I** Boxplot depicting changes in gene expression in PDXs treated with PT2399 (data from ref. 26 including sensitive PDXs only, n = 12 biological replicate PDX tumors per condition) for genes grouped

by whether their expression in 786 O cells is decreased by *shARNT* and not by *shBMAL1* (ARNTsp, yellow, n = 1069 genes), by *shBMAL1* and not by *shARNT* (BMAL1sp, purple, n = 1707 genes), by either *shARNT* or *shBMAL1* (Overlap, salmon, n = 1273 genes), or neither (NA, gray, n = 15,584 genes). **** P = $3.27e^{-10}$ by two-way ANOVA with Tukey's correction. Boxes depict the median and interquartile range (IQR), whiskers extend either to the minimum or maximum data point or 1.5 × IQR beyond the box, whichever is shorter. Outliers (values beyond the whisker) are shown as dots. **J–L** Volcano plots depicting expression changes for individual genes in groups depicted in (**I**). Genes with padj <0.05 are colored in red (fold change >1.5) or blue (fold change <0.67). padj is calculated in DESeq2 using p-values attained by the Wald test and corrected for multiple testing using the Benjamini and Hochberg method. Top non-redundant GOBP (**M**) or KEGG (**N**) pathways with ≥15 genes, FDR < 0.05, fold enrichment ≥2 enriched among ARNT-specific or BMAL1-specific target genes in 786O cells. Source data are provided as a Source Data file.

associated with BMAL1 or HIF2α, respectively. So, there is much less overlap between the BMAL1 and HIF2α cistromes when BMAL1 expression is reduced. Although MACS2 indicated several peaks bound to HIF2α exclusively in BMAL1-depleted 786O cells, visual inspection and motif enrichment analyses do not support widespread redistribution of HIF2α to novel loci in BMAL1-depleted cells (Fig. 5B and Supplementary Fig. S8C–E). Instead, HIF2α seems to be absent from a subset of its target loci in BMAL1-depleted 786O cells and its association with other genomic regions is preserved. We measured the chromatin association of BMAL1 and HIF2α in another ccRCC cell line, A498 (Supplementary Fig. S9), and observed a similar loss of HIF2α recruitment upon depletion of BMAL1. Together, these findings demonstrate that BMAL1 plays an important role in determining genomic localization of HIF2α in ccRCC cells.

### BMAL1-dependent HIF2α recruitment is associated with BMAL1-dependent expression

By integrating CUT&RUN results with RNA sequencing, we probed how BMAL1 and HIF2α occupancy are related to the impact of depleting ARNT or BMAL1 on gene expression (Fig. 6A, B). In 786 O cells, the 336 peaks that are co-occupied by BMAL1 and HIF2α are associated with 296 genes, 208 of which are expressed in 786O cells. Of the 208 expressed genes co-occupied by BMAL1 and HIF2α in 786O cells, 142 (68%) were differentially expressed upon shRNA-mediated depletion of ARNT or BMAL1. Of those, 101 (71%) exhibited significantly decreased expression, consistent with the hypothesis that ARNT and/ or BMAL1 support HIF2α-driven expression of those genes in 786O cells. Notably, 41 of those genes were decreased by depletion of either ARNT or BMAL1. Genes occupied by BMAL1 but not by HIF2α were more likely to be decreased in cells expressing *shBMAL1*, while genes occupied by HIF2α but not by BMAL1 were more likely to be decreased in cells expressing *shARNT*. A surprisingly large number of genes occupied by BMAL1 and/or HIF2α exhibited reduced expression upon depletion of either BMAL1 or of ARNT.

Focusing on how depletion of BMAL1 affects HIF2α occupancy (Fig. 6C, D), we found that genes associated with BMAL1-dependent HIF2α occupancy tend to exhibit reduced expression upon depletion of BMAL1. Of the 1207 peaks identified in HIF2α-associated chromatin in 786O cells, 509 (42%) were not identified as occupied by HIF2α in BMAL1-depleted cells (Supplementary Fig. S8C). Those 509 peaks are associated with 395 genes, of which 261 are expressed in 786O cells. Of those, 162 (62%) exhibited altered expression upon shRNA-mediated depletion of BMAL1 or ARNT, including 85 and 73 that were significantly decreased by *shBMAL1* and *shARNT*, respectively. Transcripts associated with HIF2α-occupied peaks that were retained in cells expressing *shBMAL1* were less likely to be affected by *shBMAL1* and more likely to be reduced in ARNT-depleted cells.

Finally, genes near chromatin loci bound to both BMAL1 and HIF2α that exhibited significantly altered RNA expression in BMAL1-

depleted cells are enriched in pathways related to metabolic functions. Genes that are associated with HIF2α in control cells and exhibit enhanced expression upon BMAL1 depletion are enriched in pathways related to angiogenesis (Fig. 6E). Together, these findings support the idea that BMAL1 and ARNT promote the expression of overlapping and distinct HIF2α-dependent gene networks and suggest that BMAL1 may preferentially promote expression of HIF2α target genes that impact metabolism.

### BMAL1-HIF2α heterodimers are disrupted by HIF2α antagonists

Based on the critical requirement for HIF2α to drive the formation and growth of ccRCC, elegant work led to the development of HIF2α antagonists that disrupt ARNT-HIF2α heterodimers by interacting with a surface pocket in the PAS-B domain of HIF2α[26]. HIF2α antagonists like PT2399 disrupt ARNT-HIF2α by causing a conformational change in HIF2α PAS-B, resulting in a clash between a methionine (M252) in HIF2α and a glutamine (Q447) in ARNT[41,42]. BMAL1 contains a similarly bulky amino acid (M423) in the analogous loop of the PAS-B domain (Supplementary Fig. S10A), suggesting that BMAL1-HIF2α heterodimers would also be disrupted by HIF2α antagonists. To evaluate whether BMAL1-HIF2α is disrupted by HIF2α antagonists, we expressed a normoxia-stabilized HIF2α mutant with FLAG-tagged ARNT, ARNT2, BMAL1, or BMAL2 in HEK293 cells. Purification of FLAG-tagged proteins revealed that the interactions of HIF2α with each of these partners are disrupted by PT2399, with BMAL1-HIF2α appearing to be more readily disrupted than ARNT-HIF2α (Fig. 7A,B and Supplementary Fig. S10B). To quantitatively compare the impact of PT2399 on the transactivation activities of ARNT-HIF2α and BMAL1-HIF2α, we turned to luciferase reporter assays. Using this approach, we find that expression of HRE-driven luciferase is more sensitive to suppression by PT2399 in cells overexpressing stabilized HIF2α in combination with BMAL1 than it is in cells in which stabilized HIF2α is combined with overexpression of ARNT (Fig. 7C).

HIF2α antagonists are effective at reducing the growth of many ccRCC tumors, but resistance to these drugs in up to 30% of cases remains unexplained[14,26,43]. We analyzed publicly available RNA sequencing data from a study that investigated differences between patient-derived xenograft tumors that were either sensitive or resistant to growth suppression by HIF2α antagonists[26]. *BMAL1* mRNA expression was higher in patient-derived xenografts that were sensitive to growth suppression by PT2399, while *ARNT* was unchanged (Fig. 7D and Supplementary Fig. S10C, D). Notably, the expression of BMAL1-CLOCK target genes (e.g., *NR1D1*) was increased by PT2399 treatment in sensitive tumorgrafts but not in those that were resistant to growth suppression by the drug (Fig. 7D and Supplementary Fig. S10E), suggesting a shift from BMAL1-HIF2α heterodimers towards BMAL1-CLOCK in cells that respond to the antagonist. As discussed earlier, expression of transcripts that we found to be dependent on BMAL1 (e.g., *SLC2A14*) or on ARNT (e.g., *ADM*) in 786O cells were

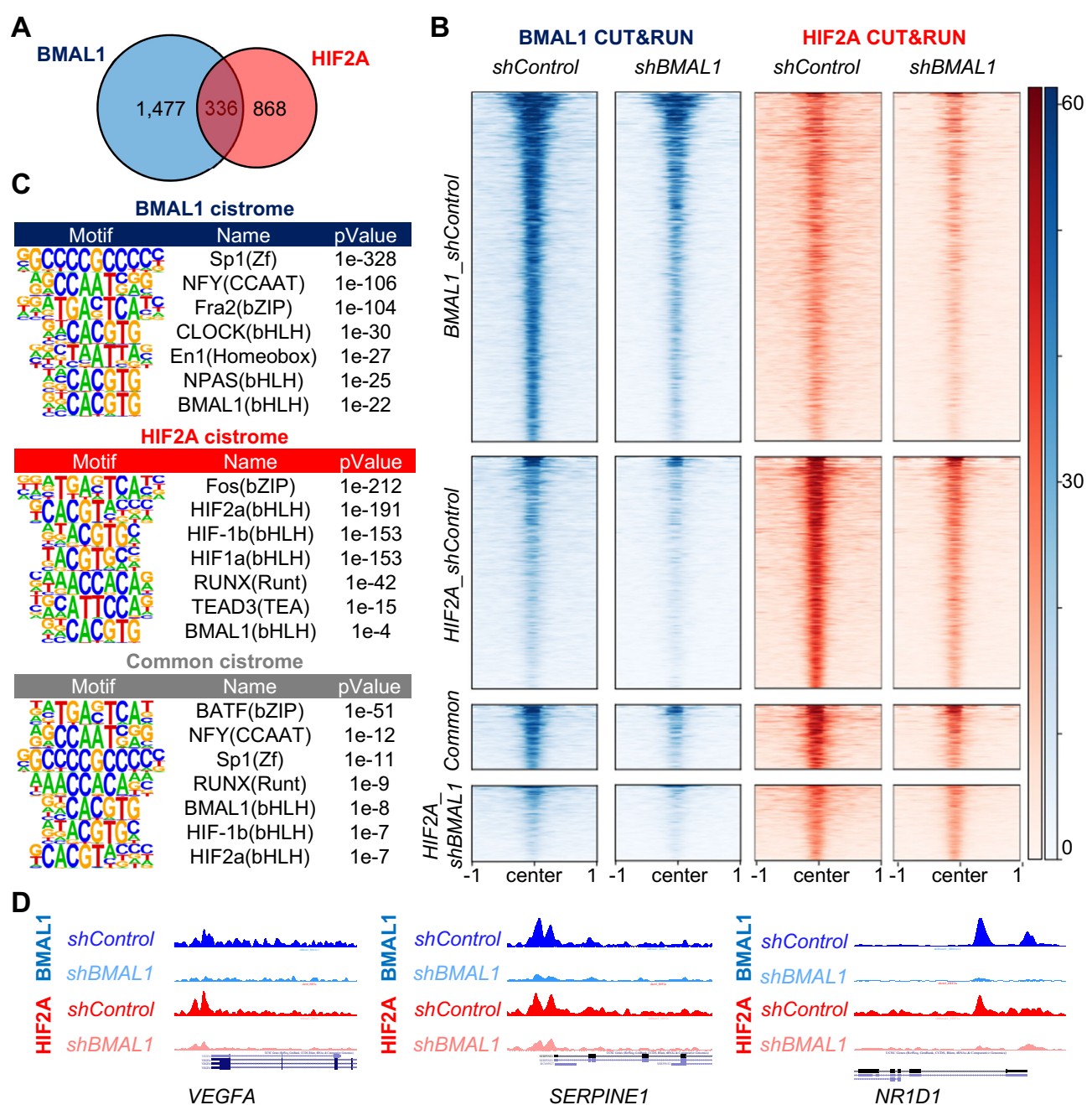

**Fig. 5 | BMAL1 influences recruitment of HIF2α to a subset of target genes.**
**A** Venn diagram depicting the numbers of genomic sites ("peaks") identified in chromatin fragments isolated by CUT&RUN procedure from 786O cells using antibodies recognizing BMAL1 (blue) or HIF2α (red). n = 3 samples per condition. **B** Chromatin binding profiles of BMAL1 and HIF2α in CUT&RUN samples (n = 3 per condition) prepared from 786O cells expressing the indicated shRNAs. Peaks are depicted in four clusters: BMAL1 peaks in 786O cells expressing *shControl* (top cluster: 1813 peaks), HIF2α peaks in 786 O cells expressing *shControl* (second cluster: 1207 peaks), peaks associated with both BMAL1 and HIF2α in 786 O cells

expressing *shControl* (third cluster: 336 peaks), or HIF2α peaks identified only in 786 O cells expressing *shBMAL1* (bottom cluster: 393 peaks). **C** Transcription factor binding motifs enriched in chromatin associated with BMAL1 (blue), HIF2α (red), or both (common, gray) in *shControl* cells. p-values were calculated in HOMER based on the hypergeometric distribution as described in ref. [40]. **D** Representative genome browser tracks for BMAL1 and HIF2α CUT&RUN in 786 O cells expressing *shControl* or *shBMAL1*, showing peaks in *VEGFA*, *SERPINE1*, and *NR1D1* loci. Data represent merged read counts for triplicate samples for each condition. Source data are provided as a Source Data file.

decreased by PT2399 treatment in sensitive PDXs; these responses were blunted in tumorgrafts that were resistant to growth suppression by PT2399 (Fig. 7D). Next, we asked how depletion of BMAL1 in 786O cells affects transcriptional responses to PT2399. Like PT2399-sensitive PDXs, 786O cells increased expression of *NR1D1* and decreased expression of *SLC2A14* and *ADM* following PT2399 treatment. Depletion of BMAL1 abolished the induction of *NR1D1* by PT2399 and diminished its effects on *SLC2A14* and less so on *ADM* (Fig. 7E).

Together, these findings support the idea that BMAL1-HIF2α drives a distinct target gene network and is disrupted by HIF2α antagonists like PT2399. Further, they indicate that disruption of BMAL1-HIF2α by PT2399 promotes the activation of BMAL1-CLOCK target genes.

### BMAL1-HIF2α heterodimers contribute to tumor growth
The interaction surfaces between ARNT and HIF2α resemble those between BMAL1 and CLOCK. We took advantage of three-dimensional

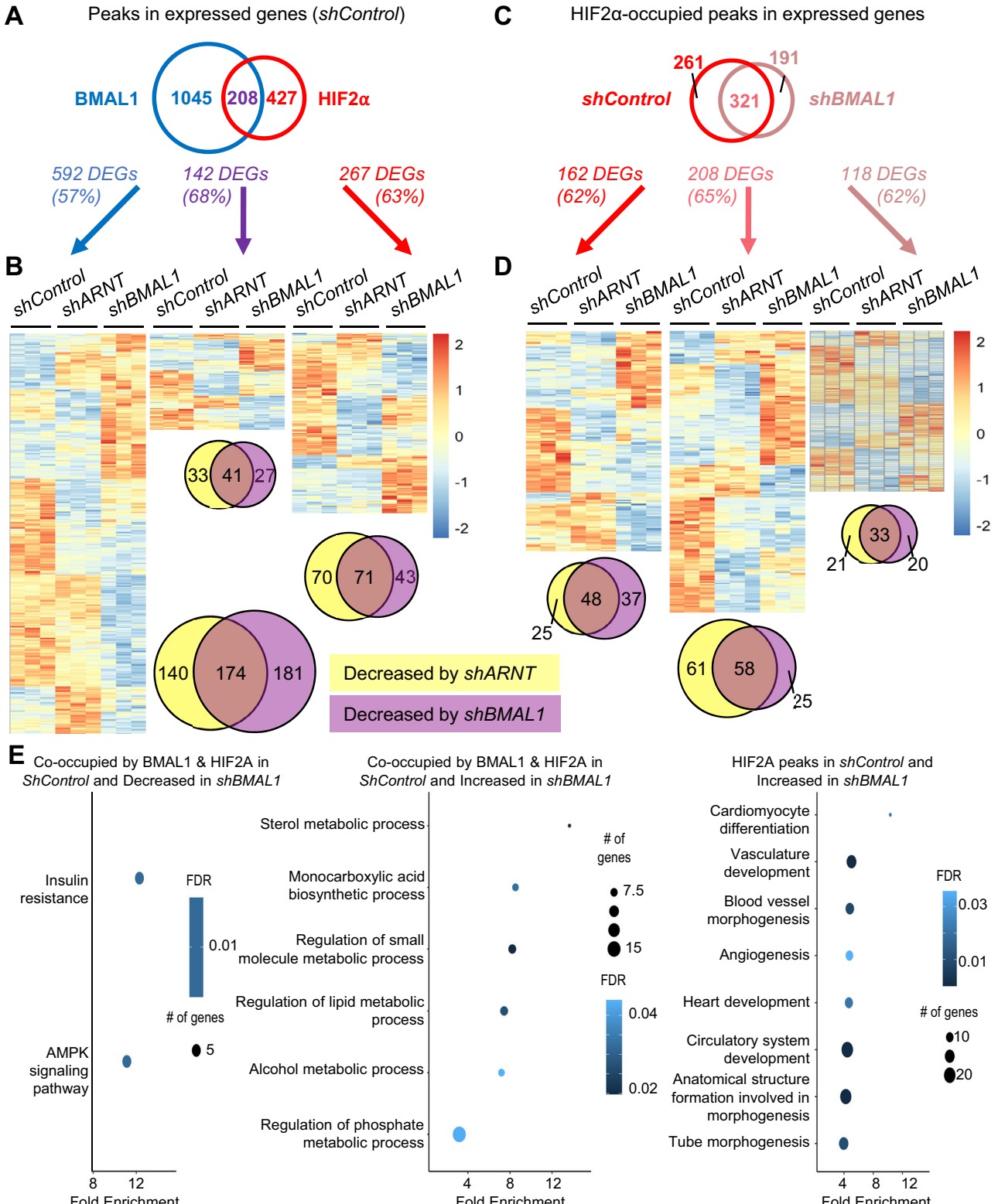

**Fig. 6 | BMAL1 chromatin occupancy correlates with BMAL1-dependent expression. A, C** Venn diagrams depicting the numbers of genomic sites ("peaks") identified in chromatin fragments isolated by CUT&RUN procedure from 786O cells using antibodies recognizing BMAL1 (blue) or HIF2α (red, pink) that are associated with genes that were detected in RNA prepared from 786O cells expressing shRNA targeting a control sequence (*shControl*) or BMAL1 (*shBMAL1*). **B, D** Heatmaps depicting differentially expressed genes (DEGs) associated with the chromatin binding sites bound by BMAL1 and/or HIF2α depicted in (**A, B**) as indicated. (FDR < 0.1 by DESeq2 for *shControl* vs. *shARNT* or *shControl* vs. *shBMAL1*). **E** Pathway enrichment in HIF2a-occupied genes grouped by BMAL1 occupancy and impact of *shBMAL1*. The x-axis represents the Enrichment Ratio, and the y-axis represents enriched GOBP pathways for genes associated with HIF2α peaks in 786O cells expressing *shControl* with at least 5 genes, FDR < 0.05, and fold enrichment >2. Expression of these genes was increased or decreased in 786O cells expressing *shBMAL1* compared to 786O cells expressing *shControl*. Source data are provided as a Source Data file.

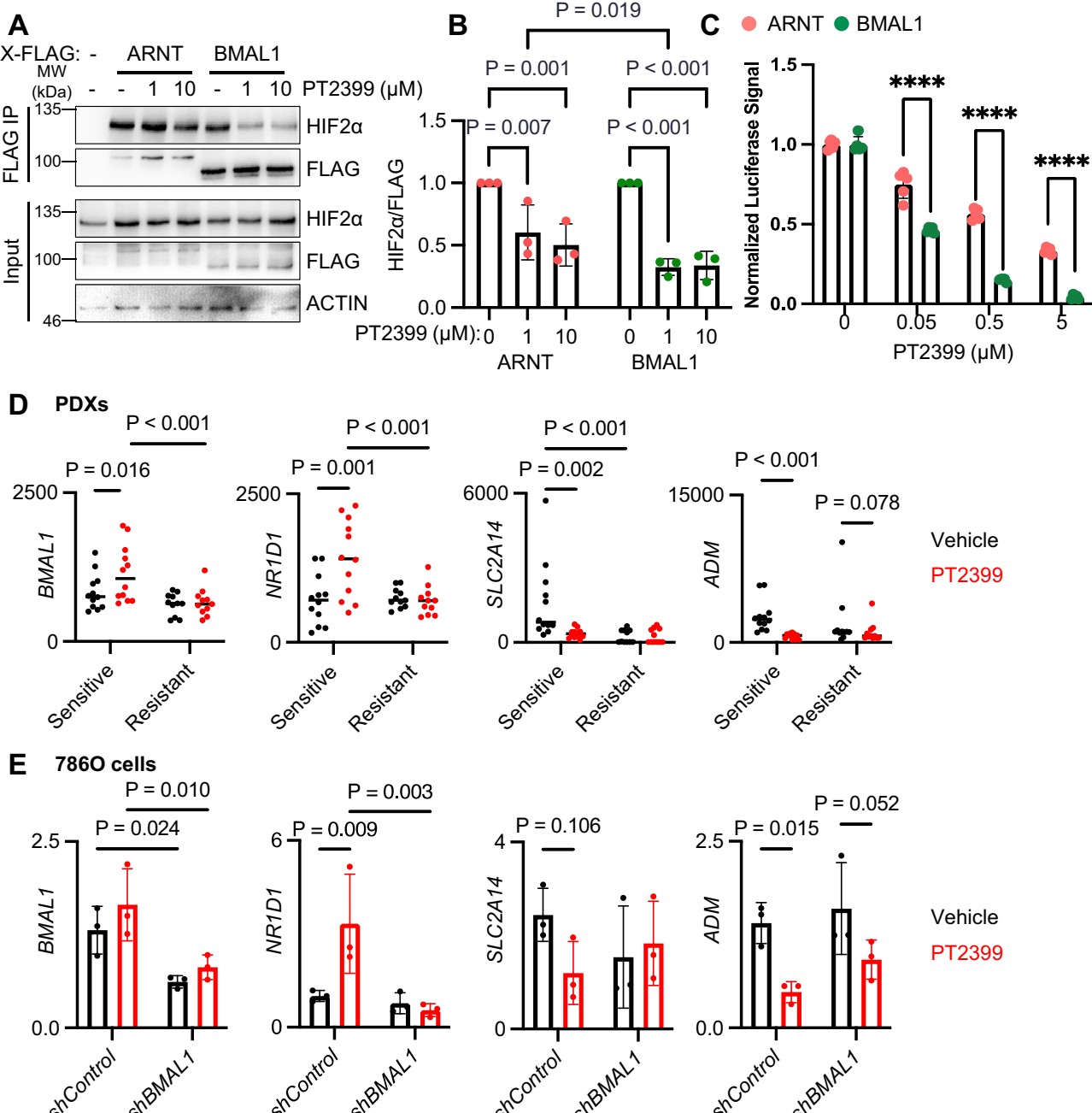

**Fig. 7 | BMAL1-HIF2α heterodimers are sensitive to disruption by PT2399.**
**A** Detection of HA-tagged stabilized HIF2α and FLAG-tagged class I bHLH-PAS proteins by immunoblot of lysates (input) or complexes purified with an anti-FLAG antibody from HEK 293 cells expressing the indicated plasmids and treated with the indicated concentrations of PT2399 for 1 h. **B** Quantitation of data like that shown in (**A**), normalized to 0 μM control groups. Data represent the mean ± s.d. for n = 3 biological replicates. **C** Relative luminescence units detected in HEK293T cells expressing luciferase under the control of a hypoxia-responsive element with overexpressed stabilized HIF2α and ARNT (salmon) or BMAL1 (green) and treated with the indicated concentrations of PT2399. Data show mean ± s.d. for n = 5 bio-logical replicates from one of three experiments with similar results. **** P < 1e⁻⁶. **D,**

**E** Detection of *BMAL1*, *NR1D1*, *SLC2A14*, and *DDIT4* transcripts in patient-derived xenograft tumors collected from mice treated with PT2399 (red) or vehicle control (black) grouped by whether tumor growth was reduced by PT2399 (sensitive) or not (resistant) (**D**) or in 786O cells expressing the indicated shRNAs and treated with 10 μM PT2399 (red) or vehicle control (black) for 6 h (**E**). Data represent transcripts per million detected by sequencing RNA collected from 11 resistant or 12 sensitive PDX tumors (**D**) or the mean ± s.e.m. for relative expression normalized to *RPLPO* by quantitative RT-PCR for three biological replicates each measured in triplicate (**E**). *P < 0.05, **P < 0.01, ***P < 0.001, ****P < 0.0001 by two-way ANOVA or mixed effects analysis with Tukey's correction for multiple hypothesis testing. Source data are provided as a Source Data file.

structures for each of these heterodimers to generate a point mutation in BMAL1 (D144A) that disrupts BMAL1-HIF2α but not BMAL1-CLOCK (Supplementary Fig. S11). To investigate whether elevated BMAL1 expression can promote the growth of ccRCC xenograft tumors in vivo, we used lentivirus to overexpress BMAL1 in 786O cells before

implanting them in immunocompromised mice. Overexpression of BMAL1 robustly increased the growth of xenograft tumors, while overexpression of BMAL1 D144A increased tumor growth to a much lesser extent (Fig. 8A and Supplementary Fig. S11E). This indicates that BMAL1-HIF2α heterodimers primarily drive the increase caused by

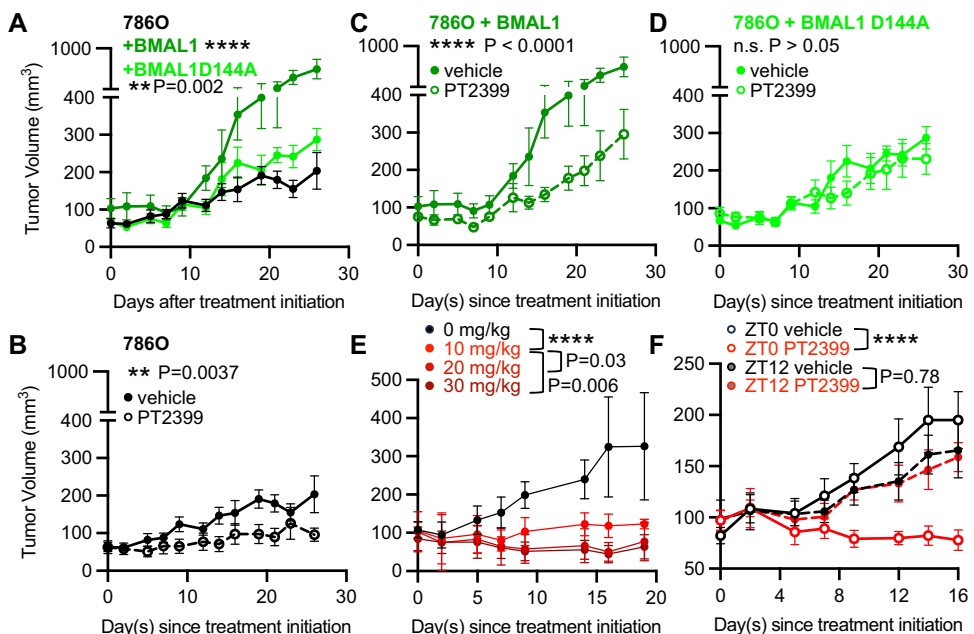

**Fig. 8 | BMAL1 promotes ccRCC xenograft growth and sensitivity to PT2399 in vivo. A–F** Volume of xenograft tumors grown from 786O cells implanted in flanks of female NIH-III Nude mice treated with 10 mg/kg of PT2399 unless otherwise indicated or vehicle control three times weekly by oral gavage between ZT0 and ZT2 (**A–E**) or at ZT0 (open symbols) or ZT12 (filled symbols) (**F**). In (**A–D**) dark and light green symbols represent xenografts grown from 786O cells expressing wildtype (dark green) or D144A mutant (light green) BMAL1; filled and open symbols represent tumors in mice treated with vehicle control or with PT2399, respectively. In (**E**, **F**) black and red symbols represent xenografts grown in mice treated with vehicle control or with PT2399, respectively. **P < 0.01, ****P < 0.0001 by simple linear regression. In (**A**, **E**), P values represent comparison to 786O and to 30 mg/kg, respectively. In (**A–E**), n = 5 animals per group. In (**F**), n = 3 for the group treated with PT2399 at ZT12 because two mice died before the completion of the study. Data represent mean ± s.e.m. Source data are provided as a Source Data file.

BMAL1 overexpression. We treated mice harboring 786 O xenograft tumors with or without BMAL1 overexpression with PT2399 to test whether BMAL1-HIF2α heterodimers are sensitive to HIF2α antagonists in vivo. Intriguingly, xenograft tumors that express high levels of wild-type BMAL1 are sensitive to growth suppression by PT2399, but those expressing BMAL1 D144A, which does not interact with HIF2α, are resistant (Fig. 8B-D). Indeed, the growth of tumors that express high levels of wildtype or D144A BMAL1 was indistinguishable when both were exposed to PT2399, reinforcing the presumption that they are functionally equivalent, other than the loss of HIF2α heterodimer formation in BMAL1 D144A (Supplementary Fig. S11F). Together, these findings demonstrate that BMAL1-HIF2α heterodimers promote the growth of ccRCC xenografts and are sensitive to disruption by the HIF2α antagonist PT2399 in vivo.

### Xenograft tumor growth suppression by PT2399 depends on dosing time

Differences in ARNT-dependent and BMAL1-dependent HIF2α regulation of endogenous target genes and in the sensitivity of ARNT-HIF2α and BMAL1-HIF2α heterodimers to disruption by HIF2α antagonist drugs suggest that the relative expression of ARNT and BMAL1 could influence the tumor growth suppression response to HIF2α antagonists. Based on our observation that BMAL1-HIF2α heterodimers are more sensitive to suppression by the HIF2α antagonist PT2399 than ARNT-HIF2α heterodimers are, we predicted that suppression of tumor growth by PT2399 would be greater at the time of day when BMAL1 is more active. To test this, we first established a dose of PT2399 that produces a sub-maximal response when delivered three times per week by oral gavage (Fig. 8E). We then treated mice harboring xenograft tumors initiated by 786O cells three times per week at each of two zeitgeber times (ZTs, defined as the number of hours after the lights are turned on each day). We delivered PT2399 either at the time of the dark-to-light transition (ZT0, when we predict high BMAL1

transcriptional activity[38]; Supplementary Fig. S12) or at the light-to-dark transition (ZT12, when BMAL1 transcriptional activity is expected to be low). We found that PT2399 delivered by oral gavage at ZT0 was highly effective at suppressing tumor growth while the same dose delivered at ZT12 was ineffective (Fig. 8F and Supplementary Fig. S13). This suggests that circadian rhythms could dramatically influence the effectiveness of HIF2α antagonists in ccRCC, either due to tumor-intrinsic expression of BMAL1 or other physiological rhythms.

To gain insight into the mechanisms by which treatment time influences the impact of PT2399 on xenograft tumor growth, we sequenced RNA prepared from xenograft tumors collected from mice treated with vehicle or PT2399 at ZT0 or ZT12. Using DESeq2, we identified 1,241 transcripts that were significantly altered by PT2399, treating ZT as a confounding factor (Supplementary Fig. S14A). Many of the transcripts that were altered by PT2399 responded similarly to treatment at ZT0 or ZT12, but the effect was generally more robust at ZT0, particularly for transcripts that were enhanced by PT2399 (Supplementary Fig. S14B). BMAL1-CLOCK target genes (e.g., *NR1D1*) were highly expressed at ZT0 and were induced by PT2399 treatment at ZT0 but not at ZT12, demonstrating that BMAL1 is active at ZT0 as predicted (Supplementary Fig. S14C). Notably, several genes in the Hallmark HYPOXIA and GLYCOLYSIS gene sets were more robustly affected by PT2399 treatment at ZT0 than by the same treatment at ZT12 (Supplementary Fig. S14D–F). Together, these findings suggest that BMAL1-dependent circadian rhythms could influence the sensitivity of ccRCC to HIF2α antagonist drugs.

## Discussion

The circadian transcription factor BMAL1 is closely related to ARNT, the canonical partner for HIF alpha subunits. We demonstrate that BMAL1 directly participates in HIF2α target gene regulation and promotes growth in ccRCC-derived cells and xenograft tumors. Interaction between BMAL1 and a highland-adapted variant of HIF2α

influences circadian rhythms in a Tibetan rodent[44], and BMAL1-HIF2α heterodimers can contribute to circadian rhythms in myocardial injury[45]. Our findings reported herein provide additional support for the idea that BMAL1 is an important partner in HIF2α-driven gene regulation and show that this is relevant to ccRCC, in which HIF2α is a key driver. Circadian disruption enhances the risk of several cancer types[46] and deletion of *BMAL1* has been used to study circadian disruption genetically, with mixed results in mouse models of cancer[2–4,47]. The reasons for diverse impacts of *BMAL1* deletion in different tumor types are unclear. Our findings suggest that decreased expression of HIF target genes could contribute to reduced growth upon *BMAL1* deletion in some tumor types.

Two additional homologs, ARNT2 and BMAL2, could participate in HIF2α signaling in a similar manner. We found that *ARNT2* expression is significantly reduced in ccRCC samples compared to adjacent normal kidney tissue (Supplementary Fig. S1B). A pre-publication report indicates that BMAL2 supports hypoxic responses in a pancreatic cancer model[48], and it may also play a role in ccRCC. Further investigation is needed to understand the contributions of diverse bHLH-PAS partners to HIF2α activities and responses to HIF2α antagonist drugs in diverse physiological and pathological contexts. Notably, we found that rhythmic accumulation of BMAL1 and HIF2α protein is approximately antiphase in 786 O cells. This antiphase expression means that the relative proportion of BMAL1-HIF2α heterodimers compared to ARNT-HIF2α heterodimers is much greater at the time of day when BMAL1 is high and HIF2α is low. This likely contributes to the unexpectedly large effect of time of day observed for the suppression of xenograft tumor growth by PT2399 in vivo.

HIF1α and HIF2α activate gene expression through HREs and regulate overlapping and distinct sets of target genes. Differences in their transcriptional targets are presumed to underlie the greater dependence of ccRCC on HIF2α activity, but the determinants of differential specificity are unclear. Depletion of either *ARNT* or *BMAL1* in ccRCC-derived cell lines dramatically altered gene expression, including that of hypoxia target genes. Some HIF targets were reduced upon depletion of *BMAL1*, and others were enhanced, suggesting that subsets of genes are preferentially activated by distinct HIF2α-containing heterodimers. This possibility is further supported by enrichment of overlapping and distinct nucleotide sequences in chromatin purified with BMAL1 or HIF2α from ccRCC-derived cells. Taken together, our findings support a mechanistic model in which the BMAL1-HIF2α heterodimer executes an overlapping but different program related to that executed by the ARNT-HIF2α heterodimer.

Despite the divergent impacts on gene expression of depleting ARNT or BMAL1 in ccRCC patient-derived cell lines, losing either of these HIF2α partners dramatically reduces the ability of several ccRCC cell lines to form colonies in vitro and xenograft tumors in vivo. Additional investigation will be required to determine whether specific HIF2α target genes critical for tumor growth require both ARNT and BMAL1 to reach an expression threshold that is needed to support tumor formation or if loss of distinct genes driven by ARNT-HIF2α or by BMAL1-HIF2α contributes to growth impairment upon depletion of each heterodimer. Notably, direct interaction between two BMAL1-CLOCK heterodimers and histones can promote gene expression via tandem E-boxes and the BMAL1-CLOCK heterodimer was shown to compete with histones for DNA access[22]. These observations suggest mechanisms by which multiple bHLH-PAS heterodimers could cooperatively influence the expression of common target genes. Additional research is needed to determine how ARNT and BMAL1 cooperate to support HIF2α activities in ccRCC.

Approximately 30% of ccRCC patient-derived xenograft tumors are resistant to HIF2α antagonists, with resistant tumors exhibiting no significant changes in gene expression following treatment[26]. Thus, the sensitivity of ccRCC to treatment with HIF2α antagonists is associated with changes in gene expression; and such sensitivity has been shown to require HIF2α[28]. There is currently a lack of comprehensive understanding regarding mechanisms underlying resistance to HIF2α antagonists[14,26–28]. Here, we showed that *BMAL1* expression is higher in PDXs that were sensitive to growth inhibition by PT2399 and that HIF2α-BMAL1 heterodimers are more sensitive to suppression by PT2399 than HIF2α-ARNT heterodimers are. We defined groups of genes as ARNT-specific or BMAL1-specific by RNA sequencing of ccRCC patient-derived cell lines in which ARNT or BMAL1 is depleted by shRNA. Expression for both groups was reduced by PT2399 treatment in patient-derived xenografts in which PT2399 is effective at suppressing in vivo tumor growth, further supporting the idea that ARNT and BMAL1 promote the expression of overlapping and distinct sets of HIF2α target genes that are relevant for therapeutic responses to HIF2α antagonists in ccRCC.

Finally, we demonstrated that xenograft tumor growth suppression by the HIF2α antagonist PT2399 is reduced by expression of a mutant BMAL1 that does not interact with HIF2α and is highly dependent on the time of day at which PT2399 is delivered, suggesting that BMAL1 expression and/or physiological circadian rhythms could contribute to differential sensitivity to HIF2α antagonist drugs in ccRCC. We and others have shown that drug transport and metabolism are influenced by circadian rhythms[49,50], so differences in the absorption, distribution, or metabolism of PT2399 may contribute to the observed differential response. We consider this unlikely because many target genes were similarly impacted by treatment at ZT0 and ZT12. The potential for documented rhythms in tissue oxygenation[51] to influence transcriptional responses of HIF2α at different times of day will require additional investigation. Together, the findings reported here suggest that BMAL1 enhances sensitivity to HIF2α antagonists through the formation of a BMAL1-HIF2α heterodimer that is more sensitive to suppression by HIF2α PAS-B domain ligands.

## Methods

Our research complies with all relevant ethical regulations. Animal care and treatments were approved by the Scripps Research Institutional Animal Care and Use Committee under protocol #10-0019. Experiments involving recombinant DNA, viruses, human cell lines, and other potential biohazards were approved by the Scripps Research Institutional Biosafety Committee under protocol #11-10-10-17.

### Analyses of RNA sequencing data from TCGA projects

RNA sequencing data for five projects in The Cancer Genome Atlas (TCGA) and from the clinical proteomic tumor analysis consortium 3 (CPTAC3) were downloaded from the NIH genome data commons (https://portal.gdc.cancer.gov/). Expression of *ARNT*, *ARNT2*, *BMAL1*, and *BMAL2* was extracted, analyzed, and visualized in Rstudio using packages rstatix and ggpubr. Clock correlation distance analysis was performed using the online tool available through the Hughey lab (https://hugheylab.shinyapps.io/deltaccd/). Software used for statistical analysis and data visualization are available via GitHub (see Code Availability statement). The number of samples included in each group are: BRCA (113 normal; 1111 tumor), COAD (82 normal; 962 tumors), KIRC (72 normal; 541 tumors), KIRP (100 normal; 872 tumors), LUAD (59 normal; 539 tumors).

### Cancer dependency map analysis

Dependency data (DepMap_Public_23Q4+Score,_Chronos) for 37 kidney cell lines were downloaded from the Cancer Dependency Map portal (https://depmap.org/portal/) on February 29, 2024. Statistical analysis and data visualization were performed in RStudio using packages rstatix and ggpubr. The code is available at GitHub.

### Immunohistochemistry for BMAL1 in human tissue samples

Immunohistochemistry was performed on a kidney cancer tumor microarray (KD2001; tissuearray.com). This slide includes 138 ccRCC

core samples, 4 clear cell papillary renal cell carcinoma core samples, 4 sarcomatoid carcinoma core samples, 4 Papillary renal cell carcinoma (II type) core samples, 4 Chromophobe carcinoma core samples, 26 invasive low grade urothelial carcinoma core samples, and 20 normal kidney core samples. Slide was deparaffinized in xylene and rehydrated through a graded alcohol series (100%, 95%, and 70% ethanol). Antigen retrieval was performed using citrate buffer (pH 6.0) at 95 °C for 15 min. Endogenous peroxidase activity was blocked by incubation with 3% $H_2O_2$ at room temperature for 10 min. Slide was blocked for 1 h with 2.5% normal horse serum (Vector Laboratories) and then incubated overnight at 4 °C with an anti-BMAL1 antibody (Novus #NB100-2288). The next day, the slide was incubated with ImmPRESS DAB reagent (Vector Laboratories) for 1 h, followed by development with the DAB peroxidase (HRP) substrate kit (Vector Laboratories). The slide was counterstained with hematoxylin, dehydrated, and mounted with mounting medium. The Immunostained slide was scanned using Aperio AT2 slide scanner (Leica) at ×20 magnification and analyzed with QuPath software.

### Lumicycle

A498, 786 O, WT8, and RCC4 (WT or +VHL) cells were infected with lentivirus expressing *Per2-Luciferase* in the pLenti/R4R2 vector. $2 \times 105$ cells were plated per 35 mm dish (VWR # 82050E538) and 6 plates per condition were used. The following day, cells were synchronized by adding DMEM containing 1 µM of dexamethasone or 50% horse serum and 100 µM D-luciferin to the plates for 1 h. After 1 h the media was replaced with DMEM, 5% FBS, 1% penicillin-streptomycin, 15 mM Hepes, pH 7.6, and 100 µM D-luciferin. Plates were sealed with vacuum grease (VWR # 59344-055) and 40 mm glass cover slips (Thermo Scientific # 22038999) and then placed in the LumiCycle to monitor bioluminescence for the indicated number of days. LumiCycle Analysis software (Actimetrics, Inc.) was used to calculate the period and amplitude of the recorded data, which was calculated using an LM fit (sin) wave based on the running average fit for each plate.

### Treatments used for circadian synchronization in cell culture

786O cells were seeded at a density of $1 \times 10^6$ cells/mL into 10 cm plates. The following day, cells were synchronized with DMEM containing 1 µM dexamethasone for 1 h. After 1 h, the media was replaced with DMEM containing 5% FBS and 1% penicillin-streptomycin. Cells were lysed at indicated times using RIPA buffer supplemented with protease (Thermo Scientific #A32953) and phosphatase inhibitors (Sigma #P5266 and #P0044). Protein levels were quantified using the Pierce BCA Protein Assay Kit (Thermo Fisher #PI23225) and equilibrated to 1 mg/mL before proteins were visualized via western blot.

### Cell culture

786-O (ATCC® CRL-1932™), A-498 (ATCC® HTB-44™), HEK293T (ATCC® CRL3216™), and U2OS (ATCC® HTB-96™) cells were purchased from the American Type Culture Collection. 786-O (CRISPR-control), WT8, and RCC4 cells were provided by Dr. Celeste Simon. All cell lines were cultured in Dulbecco's modified Eagle's medium + 10% fetal bovine serum (Thermo Fisher Scientific) and 1% penicillin-streptomycin (Gibco), and maintained in an atmosphere containing 5% $CO_2$ at 37 °C.

### Generation of cell lines expressing shRNA

To generate cell lines expressing shRNA, lentiviral shRNA constructs encoded in PLKO.1 vectors (Sigma-Aldrich, SHC002 (*shControl*), TRCN0000003816 (*shARNT*), *shBMAL1* (a gift from Dr. Satchidananda Panda), TRCN0000019097 (independent *shBMAL1*)) were produced by transient transfection in HEK293T cells. Target cells were infected with lentivirus for 4–6 h before selection in Dulbecco's modified Eagle's medium + 10% fetal bovine serum (Thermo Fisher Scientific) and 1% penicillin-streptomycin (Gibco) containing 2.5 µg/mL puromycin for 1 week. After initial selection, cells were maintained in DMEM containing 1.25 µg/mL.

### Co-immunoprecipitation

786O cells were transfected with ARNT-FLAG, ARNT2-FLAG, BMAL1-FLAG, and BMAL2-FLAG in the pTwist CMV Hygro vector (purchased from Twist Bioscience) and pcDNA3.1-HIF2α-HA(Stb) (a gift from Dr. Carrie Partch) using the Lipofectamine® 2000 DNA Transfection Reagent Protocol. Prior to cell lysis, cells were placed in a 3% $O_2$ incubator for 4 h or kept in an incubator with 21% $O_2$. Transfections in HEK293T cells were performed in 10 cm³ tissue culture plates using polyethylenimine (PEI; Polysciences Inc #23966-2) and 2 µg of each indicated plasmid following standard protocols. Cells were treated with MG132 (10 µM) for 3 h before the addition of vehicle control (0.01% Dimethyl sulfoxide (DMSO)) or the indicated concentration of PT2399 (MedChemExpress # HY-108697) for 1 h before immunoprecipitation.

Cells were lysed using RIPA buffer supplemented with protease (Thermo Scientific #A32953) and phosphatase inhibitors (Sigma #P5266 and #P0044). Protein levels were quantified using the Pierce BCA Protein Assay Kit (Thermo Fisher #PI23225) and equilibrated before FLAG-tagged proteins were immunoprecipitated using anti-Flag M2 agarose beads (Sigma #A2220).

### Western blotting

Cell lysates were separated using 8% SDS–polyacrylamide gel (National Diagnostics #EC8901LTR) by electrophoresis (Bio-Rad #1658001) and transferred using the Trans-Blot Turbo transfer system (Bio-Rad #17001915). Proteins were detected by standard Western blotting procedures.

Primary antibodies were used for Western blotting to detect HA epitope (Sigma polyclonal #H6908, Lot 0000259082) diluted 1:1000, FLAG epitope (Sigma polyclonal #F7425) diluted 1:1000, βActin (Sigma mouse monoclonal AC-15, #A1978) diluted 1:2000, HIF2α (Novus Biologicals polyclonal #NB100-122, Lot CP) diluted 1:500, BMAL1 (Abcam polyclonal #ab93806, Lot GR1559049; and VWR monoclonal #102231-824) diluted 1:1000, HIF-1β/ARNT (Cell Signaling Technologies C15A11 Rabbit mAb #3414 Lot 2) diluted 1:1000, and KU80 (Takara # Y40400, Lot A0100015) diluted 1:1000. Secondary antibodies used were Goat Anti-Mouse IgG (H + L)-HRP Conjugate (Bio-Rad #1706516), Goat Anti-Rabbit IgG (H + L)-HRP Conjugate (Bio-Rad #1706515). All secondary antibodies were diluted 1:5000. SuperSignal West Pico PLUS Chemiluminescent Substrate (Fisher Scientific #PI34095) or Immobilon Forte Western HRP substrate (Sigma #WBLUF0500). Imaging and quantification were performed using the ChemiDoc XRS+ System (Bio-Rad #1708265) and Image Lab software version 6.1.0 build 7. Proteins detected by immunoblotting were normalized to the housekeeping protein β-ACTIN.

### Luciferase assay

HEK293T cells were seeded at a density of 15,000 cells per 96-well. Cells were transfected using standard polyethylenimine (PEI) protocols in suspension at time of seeding with 30 ng reporter HRELuc (Addgene #26731, deposited by Dr. Navdeep Chandel); 5 ng BMAL1; 15 ng HIF2α; 5 ng for ARNT; 5 ng *Renilla* Luciferase (a gift from Dr. Ian MacRae). All plasmid dilutions were prepared fresh immediately before transfection. A media change was performed on the day following transfection, at which time vehicle (DMSO) or PT2399 was added where indicated. The following day luciferase activity was measured using the Dual-Glo® Luciferase Assay System (Promega #E2920) and Infinite® 200 PRO microplate reader (TECAN #30190085).

## Protein expression and purification

Full-length human HIF2α and BMAL1, each with an N-terminal Strep tag, were each cloned into pAC8 vectors for insect cell expression. Recombinant baculoviruses were prepared in the *Spodoptera frugiperda* (sf9) cells using the Bac-to-Bac system (Life Technologies). HIF2α and BMAL1 were co-expressed in *Trichoplusia ni* Hive Five cells by infection of 25 ml each of baculoviruses per 1 L of High Five culture. Cells were harvested 48 h post-infection and lysed by sonication in a buffer containing 25 mM Tris-HCl pH 8.0, 400 mM NaCl, 5% glycerol, 0.5 mM TCEP, 1 mM MgCl$_2$, 1× protease inhibitor cocktail (Roche Applied Science), and 0.1% Triton X-100. Lysate was clarified by ultracentrifugation at $186,000 \times g$ for 30 min. The supernatant was then loaded onto a gravity column for affinity chromatography containing a Strep-Tactin Sepharose bead slurry (IBA Life Sciences). The column was washed with a high salt (1 M NaCl) buffer followed by low salt (200 mM NaCl) buffer and then eluted at 200 mM NaCl using 5 mM desthiobiotin. The eluted fractions were then diluted to 100 mM NaCl prior to application on a heparin column (GE Healthcare) and then eluted using a linear salt gradient. Finally, samples were dialyzed to no more than 150 mM NaCl and flash frozen in 5% glycerol and stored at −80 °C.

## Mass photometry

Prior to mass photometry measurements, protein dilutions were made in MP buffer (20 mM Tris-HCl pH 8.0, 100 mM KCl, and 0.5 mM TCEP). Data were acquired on a Refeyn OneMP mass photometer. Eighteen microliters of buffer were first added into the flow chamber followed by a focus calibration. Two microliters of protein solution were then added to the chamber and mixed, and movies of 60 s were acquired. Each sample was measured at least two times independently (n = 2) and Refeyn Discover 2.3 was used to process movies and analyze molecular masses, based on a standard curve created with BSA and thyroglobulin.

## RNA sequencing and analysis

RNA was extracted from 786 O, WT8, and A498 cells infected with lentivirus expressing shRNA targeting the indicated transcripts, and from flash-frozen and crushed xenograft tumors. RNA was isolated using RNeasy Mini Kit (QIAGEN #74104) and QIAshredder (QIAGEN #79654). RNA purity was assessed by Agilent 2100 Bioanalyzer and quantified by Thermo Fisher Qubit. Total RNA samples from cell lines were sent to BGI Group, Beijing, China, for library preparation and sequencing. Reads (paired-end 100 base pairs at a sequencing depth of 20 million reads per sample) were generated by BGISEQ-500. Total RNA samples from xenografts were sent to the genomics core facility at The Scripps Research Institute, La Jolla, California, for library preparation and sequencing. Reads (paired-end 75 base pairs at a sequencing depth of 20 million reads per sample) were generated by Aviti_1000). In addition, FASTQ files containing RNA sequencing data from Chen et al[26]. were retrieved from the sequence read archive. FASTQ sequencing files were aligned to the GRCh37 Homo sapiens reference genome using SeqMan NGen 17 software (https://www.dnastar.com/manuals/installation-guide). Assembly results were analyzed and counts data were exported using ArrayStar 17 (https://www.dnastar.com/manuals/installation-guide). Differential gene expression analysis (DESeq2) and gene set enrichment analysis (GSEA) were performed using the online tool Gene Pattern (https://www.genepattern.org) to generated normalized count data and identify differentially expressed genes. The RNA-seq FASTQ files were deposited to the GEO repository. GO term analysis was performed using the online tool ShinyGO (http://bioinformatics.sdstate.edu/go/). Data visualization including Venn diagrams, heat maps, volcano plots, and GO term representative plots were generated in RStudio using the packages pheatmap, venneuler, and ggplot2. Software used for data

visualization will be available via GitHub, named by the figure number and panel designation.

## Cleavage under targets & release using nuclease (CUT&RUN)

Chromatin immunoprecipitation followed by high-throughput sequencing (ChIP-seq) were performed using CUT&RUN assay kit (CST #86652) following the manufacturer protocol. 100,000 786O or A498 cells infected with lentivirus expressing shRNA targeting "scramble" or *BMAL1* were used for each reaction. The primary antibodies used for immunoprecipitation were 5 μg of rabbit mAb IgG isotype as a negative control (CST #66362, Lot 3), 2 μg of rabbit mAb tri-methyl-lys-4 (CST # C42D8, Lot 15), 1 μg of HIF-2α rabbit mAb (CST #59973, Lot 1), or 2 μg of BMAL1 rabbit mAb (CST #14020, Lot 4). Fifty picograms of spike-in control DNA (provided in kit) were added to each sample for normalization. DNA purification was performed by phenol/chloroform extraction followed by ethanol precipitation. Next-generation sequencing libraries were prepared using DNA Library Prep Kit for Illumina (CST #56795) and Multiplex Oligos for Illumina (CST #29580). Libraries were sent to BGI Group, Beijing, China, for sequencing. Reads (paired end 100 base pairs at a sequencing depth of 20 million reads per sample) were generated by DNBSEQ. The CUT&RUN-seq FASTQ files were deposited to the GEO repository.

## CUT&RUN data analysis

The bioinformatic analysis was conducted at the HPC cluster located at Helmholtz Munich. Initial processing of raw data involved quality control using Fastqc 0.12.1 from the trim galore suite 0.6.10. Subsequently, reads underwent alignment to both the human genome hg19 and the yeast genome sacCer3 using Bowtie2 2.5.3, with the following parameters: --local --very-sensitive --fr --dovetail --no-mixed -I 10 -X 700. Alignment files (SAM) were then converted to BAM format, and subjected to filtering, and duplicate reads were removed using samtools 1.6 and sambamba 1.0. Peak calling was performed using MACS2 2.2.9.1, specifying parameters --keep-dup all --max-gap 400 −p 1e-5. Post-peak calling, filtering against the hg19 blacklist was executed using bedtools 2.31.1 with the intersect option. Finally, annotation and motif analysis of the peaks were carried out using HOMER 4.11, using annotatePeaks.pl and findMotifsGenome.pl options with the human genome hg19 reference. Peak functional annotation was directly done by Homer using -go option, or with WEB-based GEne SeT AnaLysis Toolkit -WebGestalt (https://www.webgestalt.org/) to identify gene ontologies and KEGG-related pathways after crossing peaks annotation with RNA-seq data.

Spike-in normalization with the aligned reads was achieved against the yeast genome sacCer3 with deeptools 3.5.5 using bamCoverage −scaleFactor --smoothLength 60 --extendReads 150 −centerReads to produce BigWig files. Spike-in scale factor values were calculated as described in the manufacturer's protocol (CST #86652). Profiles and heatmap were obtained by using computeMatrix −referencePoint center after spike-in normalization. BigWig files were uploaded to the UCSC genome browser (https://genome-euro.ucsc.edu/index.html) and tracks were visualized against the human genome hg19.

## Colony formation assay

Cells were counted by the Oroflo Moxi Z cell counter and plated 250 cells/well. Media was changed every 2 or 3 days until colonies were visible by eye (-10–14 days on average), at which point cells were washed with PBS, fixed for 10 min with 100% methanol, and stained with 0.05% crystal violet for 20 min. After staining, plates were rinsed in water, imaged using the ChemiDoc XRS+ System (Bio-Rad), and quantified using FIJI ImageJ (DOI 10.1186/s12859-017-1934-z).

## Cell line derived xenografts

NIH-III nude mice (Charles River Laboratories; strain code 201) were implanted in each flank with $5 \times 10^6$ cells were suspended in 1:1 ratio of PBS and Matrigel (Corning #CB-40234). The final volume for injection was 100 μL. Tumors were measured by caliper and tumor volume was calculated using the formula $V = (\pi/6)(Length)(Width^2)$. Experimental termination was determined empirically when the first mouse had a tumor measuring $600\,mm^3$ at which point mice were euthanized by $CO_2$ inhalation. The maximum tumor size permitted by Scripps Research Institutional Animal Care and Use Committee under protocol #10-0019 is $1200\,mm^3$ which was not exceeded in these experiments.

In the experiments examining BMAL1 depletion, 786O or A498 cells infected with lentivirus expressing shRNA targeting "scramble" or *BMAL1* were implanted in the subcutaneous tissue of ~8 weeks old mice. An equal mix of male and female mice were used. There were 15–20 mice per experimental group. Tumors were measured weekly.

We found there was no difference in the growth of tumors based on the sex of the murine host, so for the xenograft experiments treating mice with PT2399 (MedChemExpress # HY-108697) we used only female mice. Mice were 8–12 weeks old at the time of implanting cells in the subcutaneous tissue of the flank. There were 5 mice/experimental condition. Tumors grew to ~100 mm³ before beginning treatment. PT2399 was solubilized in 10% ethanol, 30% PEG400 (MedChemExpress # HY-Y0873A), 60% water containing 0.5% methylcellulose (Thermo Scientific # 25811-5000) and 0.5% Tween 80 (VWR # BDH7781-2). Mice were treated with 10 mg/kg of PT2399 or the equivalent amount of vehicle without PT2399 (vehicle control) three times a week by oral gavage. Tumors were measured by caliper three times per week.

For the time-of-day dosing studies, 786 O cells were implanted into the flank of the mice. Mice were placed in two separate light-tight boxes set to continuous cycles of 12 h of light followed by 12 h of dark 3 days before the drug treatment began and for the remainder of the experiment. One of the light-tight boxes was set to a reverse lighting schedule so that treating mice at ZT0 and ZT12 could be performed at the same time for feasibility and to reduce variability. Tumorgrafts were collected and flash frozen 6 h after the final treatment for RNA-sequencing.

For the BMAL1 overexpression experiments, 786O cells were infected with lentivirus expressing BMAL1 (wildtype or D144A). $2 \times 10^5$ cells were implanted into the flank of the mice and tumor volume was measured 3 times a week.

## RNA extraction and quantitative real-time PCR

RNA was extracted from 786O cells infected with lentivirus expressing shRNA targeting "scramble" or *BMAL1* and treated with 0.01% DMSO (vehicle) or 10 μM PT2399 for 4 h or from 786O cells implanted in mouse flanks, collected and flash frozen at ZT0, ZT4, ZT8, ZT12, ZT16, and ZT20 35 days after tumor implantation, and homogenized using a tissue crusher. RNA was extracted using Qiazol reagent (Qiagen cat # 799306) standard protocol. RNA was quantified by NanoDrop 2000 spectrophotometer (Thermo scientific cat # ND2000). cDNA was prepared using 1 μg of RNA and 4 μl of QScript cDNA Supermix (VWR cat # 101414-106). Thermocycling conditions were 25 °C for 5 min, 42 °C for 30 min, and 85 °C for 5 min and executed using C1000 Touch Thermal Cycler (Bio-Rad cat # 1851148). cDNA was diluted 1:40 with nuclease free water and 4 μl of diluted cDNA, 5 μl of with iQ SYBR Green Supermix (Bio-Rad cat # 1708885), and 0.5 μl of each forward and reverse primers (10 μM) was used per qPCR reaction. cDNA levels were measured by Opus 384-well Real Time PCR Detection system (Bio-Rad cat # #12011452). Cycling conditions were, step 1: 95 °C for 3 min, step 2: 95 °C for 10 s, step 3: 55 °C for 10 s, step 4: 72 °C for 30 s, step 5: go to step 2 39×, step 6: 95 °C for 10 s, step 7: melt curve 65–95 °C, increments 0.5 °C for 5 s + plate read. Amplification was measured and analyzed by Bio-Rad CFX Manager 3.1. Starting quantity (SQ) as determined by the software was used for statistical analysis.

The following primers were used to detect: *hRPLPO* (Forward: GCAGCATCTACAACCCTGAAG, Reverse: CACTGGCAACATTGCGGAC), *hBmal1* (Forward: AAGGGAAGCTCACAGTCAGAT, Reverse: GGACATTGCGTTGCATGTTGG), *hNR1D1* (Forward: GCAAGGGCTTTTTCCGTCG, Reverse: GCGGACGATGGAGCAATTCT), *hSLC2A14* (Forward: CTGCTCACGAATCTCTGGTCC, Reverse: GCCTAATAGCACCGGCCATAG), *hDDIT4* (Forward: TGAGGATGAACACTTGTGTGC, Reverse: CCAACTGGCTAGGCATCAGC), *hARNT* (Forward: CTGCCAACCCCGAAATGACAT, Reverse: CGCCGCTTAATAGCCCTCTG), *hVEGFA* (Forward: AGGGCAGAATCATCACGAAGT, Reverse: AGGGTCTCGATTGGATGGCA).

## Mutagenesis

The BMAL1 D144A mutation was made in the BMAL1-FLAG pTwist CMV Hygro vector (TWIST Bioscience) and the pLX_TRC317 BMAL1 expressing vector (Addgene plasmid # 141994) using Q5 site-directed mutagenesis (NEB # E0554S). The primers used in the reaction were: forward - TTTCTATCAGCAGATGAATTGAAACACCTCATTCTC and reverse – ATTCATCTGCTGATAGAAAAGTTGGTTTGTAGTTTG).

## Statistics and reproducibility

Statistical analyses were performed using GraphPad Prism 10. Boxplots display the median and interquartile range (IQR); whiskers extend to the minimum or maximum data point or to $1.5 \times IQR$ beyond the box, whichever is shorter. Clonogenic growth assay data are presented as the mean ± standard deviation (s.d.) from 3–4 independent experiments, each using three wells per condition. Statistical significance was assessed using two-way ANOVA with Tukey's post hoc correction for multiple comparisons. Xenograft growth curve data represent two independent experiments with five mice per condition, and significance was determined using repeated-measures ANOVA. Mice were randomly assigned to treatment groups, and tumor measurements were blinded where possible. Luciferase assay data are presented as the mean ± s.d. from 3–4 independent experiments, each using six wells per condition, and were analyzed using one-way ANOVA. Unless otherwise indicated, all other statistical comparisons were performed using ANOVA, with a significance threshold of $P < 0.05$. Significance levels are denoted as follows: $^*P < 0.05$; $^{**}P < 0.01$; $^{***}P < 0.001$; $^{****}P < 0.0001$.

Differential gene expression analysis (DESeq2) and gene set enrichment analysis (GSEA) were performed using GenePattern (https://www.genepattern.org) to generate normalized count data and identify differentially expressed genes. Analyses were conducted using three biological replicates per condition for cell-based assays and 3–5 tumors per condition for xenograft studies. CUT&RUN experiments included three biological replicates per condition, except for one sample (786 O shBMAL1, IP: BMAL1), for which sequencing was prematurely terminated. No other data were excluded from the analysis. Sample sizes were determined based on pilot experiments used to estimate effect sizes, where possible.

## Reporting summary

Further information on research design is available in the Nature Portfolio Reporting Summary linked to this article.

## Data availability

Source data are provided with this paper. The RNA-sequencing data generated in this study for Fig. 2 have been deposited in the Gene Expression Omnibus (GEO) database under accession code GSE269336. The RNA-sequencing data generated in this study for Fig. S4 have been deposited in the Gene Expression Omnibus (GEO) database under accession code GSE269339. The RNA-sequencing data generated in this study for Fig. S6 have been deposited in the Gene

Expression Omnibus (GEO) database under accession code GSE269340. The RNA-sequencing data generated in this study for Fig. S14 have been deposited in the Gene Expression Omnibus (GEO) database under accession code GSE290779. The CUT&RUN ChIP-sequencing data generated in this study for Figs. 5–6 have been deposited in the Gene Expression Omnibus (GEO) database under accession code GSE269334. The CUT&RUN ChIP-sequencing data generated in this study for Figs. S8-S9 have been deposited in the Gene Expression Omnibus (GEO) database under accession code GSE290669. All other data generated in this study are provided in the Source Data file. Source data are provided with this paper.

## Code availability

Code used to generate Figs. 1C, 2A, B, 4C, D, E, I, L6B, D, and Supplementary Figs. S1–S4, S6, and S14 is available in GitHub repository KatjaLamia/Mello_BMAL1_HIF2A (https://doi.org/10.5281/zenodo.15467805).

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

## Acknowledgements

Data used for analyses described in this manuscript were obtained from: the GTEx Portal on 10/30/2023, the sequence read archive (accession number SRP073253), and data generated by the TCGA Research Network (https://www.cancer.gov/tcga). We thank Marie Pariollaud, Megan Vaughan, Fania Feby Ramadhani, Fabiana Quagliarini, Ben Cravatt, Reuben Shaw, Michael Bollong, Xiaohua Wu, and Luke Wiseman for helpful discussions, Lara Ibrahim for assistance retrieving published RNA sequencing data and aligning to the human genome, the Scripps Department of Animal Resources for assistance with oral gavage drug treatments, and Judy Valecko for administrative assistance. K.A.L. is supported by National Institutes of Health grants R01CA211187 and R01CA271500 and research reported in this publication was supported by a pilot study award to K.A.L. from the National Center for Advancing Translational Sciences of the National Institutes of Health under Award Number UL1TR002550. D.G.C. received funding from the National Institutes of Health training grant DK136780. D.A.V. received funding from the DAAD (German Academic Exchange Service) in the context of the Helmholtz Research School for Diabetes, C.J. was supported by DFG (German Research Foundation) TRR333 BATenergy (450149205), and N.H.U. received funding from the DFG TRR333 BATenergy (450149205).

## Author contributions

Conceptualization: R.M., K.A.L. Methodology: R.M., D.G.C., S.W., C.J., M.C.S., K.A.L., C.S., N.T., N.H.U. Investigation: R.M., D.G.C., S.W., K.A.L., C.S., D.A. Visualization: R.M., K.A.L., C.S., D.A. Resources: M.C.S. Funding acquisition: K.A.L., N.T., N.H.U. Project administration: K.A.L., N.T., N.H.U. Supervision: K.A.L., N.T., N.H.U., C.J. Writing—original draft: K.A.L., C.S. Writing—review and editing: R.M., D.G.C., S.W., M.C.S., K.A.L., C.S., N.T., D.A., C.J., N.H.U.

## Competing interests

The authors declare no competing interests.
