## [Transparent Peer Review file · Nature Communications]

BMAL1 and ARNT enable circadian HIF2 α responses in clear cell renal cell carcinoma

Corresponding Author: Dr Katja Lamia

Version 0:

Reviewer comments:

Reviewer #1

(Remarks to the Author)

The clock gene BMAL1 is upregulated in ccRCC compared to healthy kidneys and is closely related to ARNT. BMAL1-HIF2 α regulates a subset of HIF2 α target genes in ccRCC, with BMAL1 depletion reprogramming HIF2 α chromatin association, reducing ccRCC growth, and increasing sensitivity to the HIF2 α antagonist PT2399.

However, this paper has several issues that need to be addressed:

1. Lack of in vivo data to validate the function of BMAL1-HIF2 α .
2. Many in vitro experiments were conducted with only one cell line.
3. BMAL1 is a clock gene, but no data in the paper addresses circadian aspects.
4. It is unclear whether BMAL1-HIF2 α is associated with the prognosis of ccRCC.
5. There is a lack of data from clinical samples of ccRCC.

All in all, this manuscript is not suitable for publication in this journal

Reviewer #2

(Remarks to the Author)

Summary:

Mello et. al's paper BMAL1-HIF2 α heterodimers contribute to ccRCC examines the capability of BMAL1 to heterodimerize with HIF2a and the consequences of interaction in ccRCC cell and tumor growth. A direct interaction between members of the clock and hypoxia pathways has been hypothesized to occur due to the molecular similarities between family members, but thorough testing of these interactions in cancer has not been performed, making this topic of importance to the field. Using RNA-sequencing, the authors showed overlapping differentially expressed genes following loss of BMAL1 or ARNT in RCC cell lines. They also demonstrated that BMAL1 and HIF2a purified from insect cells can form a stable complex, and that these proteins have overlapping peak locations in CUT&RUN in 786O RCC cells. Finally, they found that BMAL1 knockdown in 786O and A498 cells showed reduced tumor growth in a RCC xenograft model, and that a previously developed HIF2a inhibitor is responsive to levels of BMAL1 expression. Overall, this work advances our understanding of Bhlh-PAS transcription factor interactions, with evidence that this may be important during RCC. However, a few areas should be addressed to increase the impact of the manuscript. These include an updated title which is currently too broad, additional dissection and presentation of their RNA and CUT&RUN data, as well as a stronger rationale for the use of over-expression systems and the diverse cell lines chosen for different experiments (HEK293, U2OS, RCC cells).

Major comments:

- The statement in the abstract 'circadian rhythms are unexpectedly robust' in RCC should be clarified or further supported experimentally. This could include evidence that the circadian genes are indeed changing across circadian time in RCC cells or tumors.
- It is known that overexpression of proteins can produce artificial interactions (Moriya, Mol Bio of the Cell, 2017). Fig. 1G would be strengthened through several changes: 1) through demonstration of the level of endogenous protein compared to overexpressed protein, 2) by demonstrating co-IP of canonical binding partners, e.g. BMAL1/CLOCK HIF2a/ARNT, 3) reciprocal pulldown of HIF2a and blotting for BMAL1.
- While the CUT&RUN data presented in figure 4 are clear and interesting, more thorough analyses would greatly increase the impact of this experiment. For example, it is stated that the "Depletion of BMAL1 reduced chromatin association of both

BMAL1 and HIF2 α at many sites that were occupied in control cells and reduced the number of significantly enriched loci detected in chromatin purified with BMAL1” (lines 165-166). However, the quantitative degree of change is not stated. Earlier in the paragraph, it is stated that “336 loci were identified as co-occupied by BMAL1 and HIF2 α , representing 18.5% or 27.9% of the sites associated with BMAL1 or HIF2 α , respectively” (line 154). This analysis is then expected for the data in which BMAL1 is knocked down. There should be an analogous percentage for those sites that were no longer occupied in the BMAL1 knockdown data on a global scale. This information would allow us to fully understand the degree of HIF2a and BMAL1 overlap and potential activity of the BMAL1/HIF2a heterodimer.

- In Fig. 3E, ARNT knockdown caused large decreases in expression of hypoxia response genes, demonstrating ARNT has function in 786O cells. However, the HRE luciferase experiments were performed in U2OS cells, an osteosarcoma line, which likely has little relevance to ccRCC. In Fig. 6C, overexpression of HIF2a and ARNT in U2OS cells showed no increase in HRE reporter activity. Therefore, use of the inhibitor PT2399 on the HIF2a-ARNT interaction had no effect. It is unclear how to interpret this figure since HIF2a-ARNT do not drive expression of the HRE reporter. Furthermore, it is unclear if this data can be applied to ccRCC cells where previous figures demonstrated the importance of ARNT in hypoxia signaling in 786O cells.

Minor comments:

- I suggest changing the title to be more precise and informative.
- It is unclear how statistically significant BMAL1 expression is in Fig. 6A. Please provide fold change (resistant/sensitive) and p-values (or indication of statistical significance).
- There is little combinatorial analysis of CUT&RUN and RNA-seq. The only analysis is in Fig. 4E which shows pathway enrichment. Showing changes in expression of individual genes across multiple samples through a heatmap or qPCR validation of targets such as those shown in Fig. 4D would be helpful.

Reviewer #3

(Remarks to the Author)

Reviewer #4

(Remarks to the Author)

Mello et al examine the functional interaction of BMAL1 with HIF2a. Previous reports have established the interaction between hypoxia and circadian pathways and the interaction of HIF1a with BMAL1. In this manuscript the authors provide evidence that HIF2a-BMAL1 interact and they binding to overlapping sets of chromatin sites. Specifically, they show that BMAL1 recruits HIF2a in specific loci and they outline consensus sequences of binding sites. Given the role of HIF2a in RCC, they observe that BMAL1 is expressed high in RCC and provide evidence that HIF2a-BMAL1 activate HRE promoters. They further investigated the role of BMAL1-HIF2a dimers in growth of VHL deficient human RCC cells. Knock down of BMAL1 suppresses RCC cells growth in vitro and in vivo. Finally, they argue that BMAL1 expression confers sensitivity to PT2399 drug, based on in vitro observations only.

In Figure 3A, (KD western blot data), it appears that when BMAL1 is knockdown, ARNT protein expression increased. Reciprocally, KD of ARNT induced BMAL1 expression. Is this compensation mechanism?

The experiments in Figure 5 support the idea that BMAL1 contributes to the growth of VHL deficient cells in the colony formation assay and in vivo as xenografts, but they do not provide any evidence that this relates to the sensitivity to PT2399.

- a) Does shARNT or shBMAL1 shifts the sensitivity of the lines to PT2399?
- b) In the nude mice experiments are the BMAL1 KD tumors more or less sensitive to PT2399 in vivo?
- c) In the colony formation assays it appears that ARNT is essential for 786O cell survival. However, from the Depmap data in Figure 1E, the BMAL1 shows significant dependency but not in ARNT. How do we reconcile this?

The data presented in Figure 6 are supposed to support the idea that expression of BMAL1 confers sensitivity to PT2399. I have several concerns regarding this conclusion.

- a) The compound is known to interrupt the HIF2a-ARNT interaction, nevertheless it is quite paradoxical that the HRE activity of co-expressed HIF2a-ARNT is NOT affected significantly by the compound (Fig 6C). Consistent with Figure 1H, BMAL1+HIF2a expression results in much higher HRE activity than HIF2a+ARNT, and subsequent significant reduction by PT2399. Does expression of BMAL1 affects HIF2a expression? How does the expression of ARNT compares with BMAL1 in the U2OS experiments? I am concerned that the different HRE activities induced by ARNT and BMAL1 are the result of different expression of exogenous bHLH-PAS proteins?

- b) 786-O+BMAL1 number of colonies are reduced significantly by PT2399. How does this compare with 786-O+ARNT colonies?

- c) BMAL1 knockdown suppresses the 786-O cell growth (Figure 5B and C). But why there is no effect of BMAL1

overexpression on colony formation?

d) Overall I am concerned that since BMAL1 may have a low expression in 786-O cells, these forced overexpression experiments may misrepresent the functional significance of BMAL1 with regards to HIF2a activity vis a vis to ARNT, especially if based on a single HRE reporter assay. For this reason, I think it is more appropriate for the authors to test by QRT-PCR specific HIF2a target genes, since they have a complete list of ARNTspecific, BMALspecific and common targets.

e) I may misunderstand the concept here but how is it that high expression of BMAL1 provides PT2399 sensitivity but the gene appears minimally enriched in the sensitive PDX xenografts. How about ARNT expression level in sensitive or resistance samples?

f) In Figure 6B the apparent “strong” effect of PT2399 on HIF2a-BMAL1 interaction is confounded by the fact that the FLAG-BMAL1 bait at 10microM is obviously less than the bait in the control lane. To quantify the effect and draw comparison between ARNT and BMAL1 we need triplicate experiments with densitometry measurements and normalization to bait intensity.

g) Lastly, to provide evidence that the proposed model is not artifactual or driven by HIF2a-independent program we need to test PT2399 sensitivity of a BMAL1 mutant that does not interact with HIF2a. Detailed published structure function analysis of ARNT should provide info for such a mutant.

h) What is the proposed mechanism by which BMAL1 promotes sensitivity to PT2399?

Version 1:

Reviewer comments:

Reviewer #1

(Remarks to the Author)

Overall, this revised manuscript includes substantial additional data compared to the previous version.

Reviewer #2

(Remarks to the Author)

The authors responded very well to the Reviewer’s comments and the manuscript is much improved. With the new set of experiments added, we have a few minor comments:

- In the abstract, the statement, “Depletion of BMAL1 reprograms HIF2 α chromatin association”, is unclear and the word “reprograms” should be more defined. The author’s show that 393 new HIF2a peaks emerge after BMAL1 knockdown (Fig. S8). However, the author’s claim that these peaks do not “visually” constitute new HIF2a peaks. Therefore, it is confusing if the point is that HIF2a binds to distinct regions of chromatin in the absence of BMAL1 or not. We believe use of this strong language describing the relationship between HIF2a and BMAL1 on chromatin should be softened. Another example of this language is, “BMAL1 plays an important role in directing HIF2 α to the genome.” we do not think there is sufficient evidence to claim that BMAL1 “directs” HIF2a to the genome since many new HIF2a peaks emerge upon BMAL1 knockdown

- Fig. 1H shows antiphasic protein expression of BMAL1 and HIF2a. Since this paper is on their function as a heterodimer, it seems paradoxical they would have antiphasic abundance. Can you address this point in the discussion or propose a mechanistic model?

- The rhythmic expression of HIF2a protein implies it is under circadian regulation, which is regulated either directly or indirectly by BMAL1-CLOCK. Therefore, it is necessary to show if BMAL1 knock down causes HIF2a protein abundance changes to interpret the genomic localization data in Fig 5 and S8. For example, if BMAL1 knockdown causes a decrease in HIF2a protein, then the reduction of HIF2a on chromatin in BMAL1 knockdown is likely due to non-specific abundance changes rather than BMAL1 “directing” HIF2a to certain genomic loci as is currently proposed.

- Fig S11 E and F are missing in description in the figure legend.

- Although it’s explained in the legend, it would be helpful to have an individual key for each graph of Fig. 8 since sometimes a filled dot means treated with PT2399 and sometimes it means vehicle.

Reviewer #3

(Remarks to the Author)

Reviewer #4

(Remarks to the Author)

I think the authors added crucial data to support their conclusions. The in vivo data and the uses of D144 mutant show that BMAL1 promotes tumor formation in vivo and that there is diurnal variation of the BMAL1 activity in RCC. They also toned down appropriately the statements in their discussion.

One very important detail that I think it is worth emphasizing regarding the mechanism is that the HIF2a-BMAL1 heterodimer is likely (based on their data) to execute an overlapping yet different program than the HIF2a-ARNT. In vitro some of the HIF target genes are NOT decreased by BMAL1, although their response to belzutifan is increased (Figure 7E NRD1 vs SCL or ADM, or SLC29A4 in Figure S9, if I read the data correctly).

Version 2:

Reviewer comments:

Reviewer #2

(Remarks to the Author)

The authors have adequately addressed our previous comments

Reviewer #3

(Remarks to the Author)

Response to Reviewers:

We thank the reviewers for their insightful feedback that helped us to substantially improve this manuscript. In response to themes that emerged from multiple reviewers, we added substantial new data including clinically relevant analysis of BMAL1 protein in de-identified patient samples (Fig. 1D), endogenous and additional cellular models (Figs. 1F-H, 2D-F, 4-6, S4-S9), circadian rhythm characterization (Figs. 1F-H, S2, 8F, S12-S14), and the importance of BMAL1 in sensitivity to PT2399 (Figs. 7-8, S10-S14). Notably, we made the exciting finding that the time of day at which mice harboring ccRCC xenograft tumors are treated with belzutifan has a major impact on its ability to suppress tumor growth (new Figure 8F).

Reviewer #1 (Remarks to the Author): expert in epigenetics, HIF and renal carcinoma

The clock gene BMAL1 is upregulated in ccRCC compared to healthy kidneys and is closely related to ARNT. BMAL1-HIF2 α regulates a subset of HIF2 α target genes in ccRCC, with BMAL1 depletion reprogramming HIF2 α chromatin association, reducing ccRCC growth, and increasing sensitivity to the HIF2 α antagonist PT2399. However, this paper has several issues that need to be addressed:

1. Lack of *in vivo* data to validate the function of BMAL1-HIF2 α .

We now include several types of *in vivo* data to address the importance of BMAL1-HIF2 α heterodimers in the growth of ccRCC xenograft tumors.

We demonstrated that BMAL1 plays a critical role in the growth of xenograft tumors derived from both 786O and A498 ccRCC cell lines (Figure 2F in the revised manuscript). In new data, we demonstrate that overexpression of wildtype BMAL1 in 786O cells enhances the growth of xenograft tumors (new Figure 8A,C), but overexpression of a mutant BMAL1 (D144A) that disrupts its interaction with HIF2 α is much less effective at enhancing xenograft tumor growth (new Figure 8A and Supplementary Figure S11).

Using shRNA-mediated depletion in cell culture, we found that some HIF2 α target genes are more dependent on BMAL1 while others are more dependent on ARNT (Figure 4 and Supplementary Figure S6). BMAL1 protein exhibits robust circadian oscillations of expression and activity generally and we find this is also true in ccRCC cell lines (new Figure 1H). This leads to a prediction that if BMAL1-HIF2 α and ARNT-HIF2 α heterodimers drive divergent gene networks *in vivo*, we would expect to observe expression patterns for BMAL1-dependent and ARNT-dependent HIF2 α target genes in xenograft tumors collected across the circadian cycle that reflect the temporal patterns of expression for *BMAL1* and *ARNT*. We demonstrate that *BMAL1* and *ARNT* RNA exhibit circadian rhythms in 786O cell xenograft tumors and that the BMAL1-specific HIF2 α target gene *SLC2A14* is expressed in phase with canonical circadian genes like *NR1D1* and *TEF*, peaking 4-8 hours after *BMAL1*, while *VEGFA* expression shows two

peaks that follow the peaks for both *ARNT* and *BMAL1*, consistent with its regulation by both in cell culture (new Supplementary Figure S12).

Finally, our model that *BMAL1* plays an important role in supporting HIF2 α -dependent growth and survival of ccRCC tumors *in vivo* leads to a prediction that suppression of tumor growth by HIF2 α antagonist drugs would be greater at the time of day when *BMAL1* is more abundant. To test this, we first established a dose of the HIF2 α antagonist PT2399 that produces a sub-maximal response when delivered three times per week by oral gavage (new Figure 8E). We then treated mice harboring xenograft tumors initiated by 786O cells three times per week at either the time of the dark-to-light transition (ZT0, at the start of the daily phase of high *BMAL1* transcriptional activity) or at the light-to-dark transition (ZT12, when *BMAL1* transcriptional activity is low). We found that PT2399 delivered by oral gavage at ZT0 was highly effective at suppressing tumor growth while the same dose delivered at ZT12 was ineffective (new Figure 8F). We were surprised at the size of this effect, which suggests that circadian rhythms could dramatically influence the effectiveness of HIF2 α antagonists in ccRCC, either due to tumor-intrinsic expression of *BMAL1* or other factors, which will require additional investigation that is beyond the scope of this study.

To gain insight into mechanisms by which zeitgeber time (ZT) influences the impact of PT2399 on xenograft tumor growth, we sequenced RNA prepared from xenograft tumors collected from mice treated with vehicle or PT2399 at ZT0 or ZT12. Using DESeq2, we identified 1,241 transcripts that were significantly altered by PT2399, treating ZT as a confounding factor (new Fig. S14A). Many of the transcripts that were altered by PT2399 responded similarly to treatment at ZT0 or ZT12, but the effect was generally more robust at ZT0, particularly for transcripts that were enhanced by PT2399 (new Fig. S14). Notably, several genes in the Hallmark HYPOXIA and GLYCOLYSIS gene sets were more robustly reduced by PT2399 treatment at ZT0 than by the same treatment at ZT12.

2. Many *in vitro* experiments were conducted with only one cell line.

We include the following experiments in multiple cell lines:

- a) Lumicycle assays demonstrating the presence of robust circadian rhythms in 786O, A498, and RCC4 ccRCC-derived cell lines and suppression of those rhythms by re-introduction of wildtype VHL (new Figures 1F-G and Supplementary Figure S2B).
- b) Reduced colony formation in 786O, A498, and RCC4 ccRCC cell lines in which *BMAL1* expression is depleted by shRNA (original Figure 5B, now Figure 2D).
- c) Reduced growth of xenograft tumors from 786O and A498 ccRCC cell lines in which *BMAL1* expression is depleted by shRNA (original Figure 5D, now Figure 2F).
- d) RNA sequencing in 786O and A498 ccRCC cell lines in which either *ARNT* or *BMAL1* is reduced by expression of shRNA (original Figure 3 now Figure 4, Supplementary Figures S4, S5, S6, and S7).

- e) CUT&RUN assessment of chromatin binding by BMAL1 and HIF2 α (original Figure 4, now Figure 5, and new Figure 6 and new Supplementary Figures S8 and S9).

3. BMAL1 is a clock gene, but no data in the paper addresses circadian aspects.

We agree that it is important to ask how HIF2 α activity is influenced by circadian rhythms in the context of the potential for different heterodimers to drive different downstream responses. We have added several new experiments to address circadian aspects of HIF2 α biology in ccRCC.

To expand our investigation of circadian robustness in ccRCC, we used a Lumicycle instrument to continuously record luciferase activity in several ccRCC cell lines (786O, A498, and RCC4) and found that each of them exhibits robust circadian rhythms of luciferase activity when expressing destabilized luciferase under the control of the Per2 promoter (new Figure 1F). Further, we reconstituted expression of wildtype VHL in 786O and RCC4 cells and found that re-introducing VHL reduced the amplitude of rhythmic Per2-driven luciferase (new Figures 1G and S2B). Together these findings demonstrate that ccRCC cells not only have robust circadian rhythms but that a primary driver of ccRCC development (loss of functional VHL) may enhance circadian rhythms.

We used Western blotting to characterize circadian rhythms of protein expression in 786O cells synchronized with dexamethasone (new Figure 1H). Here, we found that not only are the core circadian clock components robustly rhythmic, the accumulation of HIF2 α protein exhibits circadian rhythmicity while ARNT protein accumulation seems to be constant throughout the circadian cycle in cell culture. Notably BMAL1 and HIF2 α proteins do not peak at the same time. The observation that BMAL1 and HIF2 α protein rhythms are not synchronized suggests that the relative proportions of ARNT-HIF2 α and BMAL1-HIF2 α heterodimers likely vary throughout the circadian cycle. Importantly, this rhythmic accumulation of HIF2 α occurs in cells that lack functional VHL and under the constant conditions of cell culture. This suggests that there are VHL-independent mechanisms that modulate HIF2 α protein accumulation in a circadian manner – an interesting observation that will require additional investigation beyond the scope of this study.

We measured HIF2 α protein levels in xenograft tumors collected from NIHIII nude mice at six phases of the circadian cycle *in vivo*. HIF2 α expression is much higher in the human ccRCC cells, so we can use an antibody that recognizes the human ortholog of the DNA-binding protein Ku80 (but not the mouse ortholog) to normalize HIF2 α protein detection to the protein derived from human tumor cells in the samples. Using this approach, we found that HIF2 α protein exhibits a circadian rhythm of accumulation in xenograft tumors *in vivo*, peaking during the dark phase (new Figures S12A and S12B).

Both BMAL1 and ARNT are present in healthy mouse tissues as well as in tumor cells, and we do not have antibodies that recognize either BMAL1 or ARNT in a species-specific manner. Thus, we can't properly determine the accumulation of ARNT or

BMAL1 protein in xenograft tumor cells in vivo using this approach. Instead, we used quantitative RT-PCR with human-specific primers to measure gene expression in xenograft tumors collected at six ZTs (new Figure S12C). We observed antiphase oscillating expression of *BMAL1* and *NR1D1* as expected. We also measured the expression of HIF2 α target genes and found that the BMAL1-specific target *SLC2A14* exhibits circadian expression in phase with *NR1D1*, consistent with the idea that its expression is supported by BMAL1, while expression of *VEGFA*, a HIF2 α target that is not BMAL1-dependent in 786O cells, was less clearly circadian (Figure S12C).

4. It is unclear whether BMAL1-HIF2 α is associated with the prognosis of ccRCC.

We agree that it is unclear whether BMAL1-HIF2 α is associated with the prognosis of ccRCC. Previous dogma claimed that BMAL1 and HIF2 α do not form a physiologically relevant heterodimer. Our findings establish that this heterodimer is present in ccRCC cell lines and that BMAL1-HIF2 α heterodimers support HIF2 α -dependent gene expression and cell growth in ccRCC cell lines. These findings imply that ARNT is not the only relevant partner for HIF2 α in ccRCC, in which HIF2 α is a major driver. We do not yet know whether BMAL1-HIF2 α independently influences the prognosis of ccRCC, and that question is beyond the scope of this report. We have updated the title to better capture the message of our study.

Our data support the idea that BMAL1 is one partner for HIF2 α and that either ARNT or BMAL1 can support HIF2 α transcriptional activity. It seems that both ARNT and BMAL1 are required for optimal growth of ccRCC-derived cell lines. Data from TCGA suggest that there is no association between overall survival (and several other clinical outcomes) and copy number or expression level of either ARNT or BMAL1. Our findings suggest that the relative expression of ARNT or BMAL1 may influence the response to HIF2 α antagonist treatments, or possibly to other treatments that target pathways downstream of HIF2 α in ccRCC that may be differentially regulated by ARNT-HIF2 α vs BMAL1-HIF2 α .

5. There is a lack of data from clinical samples of ccRCC.

We now include data demonstrating that nuclear staining for endogenous BMAL1 protein is higher in ccRCC compared to non-tumor kidney tissues and other types of renal cancer using a BioMAX array (new Figure 1D and Supplementary Figure S1C).

Reviewer #2 (Remarks to the Author): expert in BMAL1, clock genes

Summary:

Mello et. al's paper BMAL1-HIF2 α heterodimers contribute to ccRCC examines the capability of BMAL1 to heterodimerize with HIF2 α and the consequences of interaction in ccRCC cell and tumor growth. A direct interaction between members of the clock and hypoxia pathways has been hypothesized to occur due to the molecular similarities between family members, but thorough testing of these interactions in cancer has not been performed, making this topic of importance to the field. Using RNA-sequencing,

the authors showed overlapping differentially expressed genes following loss of BMAL1 or ARNT in RCC cell lines. They also demonstrated that BMAL1 and HIF2a purified from insect cells can form a stable complex, and that these proteins have overlapping peak locations in CUT&RUN in 786O RCC cells. Finally, they found that BMAL1 knockdown in 786O and A498 cells showed reduced tumor growth in a RCC xenograft model, and that a previously developed HIF2a inhibitor is responsive to levels of BMAL1 expression. Overall, this work advances our understanding of Bhlh-PAS transcription factor interactions, with evidence that this may be important during RCC. However, a few areas should be addressed to increase the impact of the manuscript. These include an updated title which is currently too broad, additional dissection and presentation of their RNA and CUT&RUN data, as well as a stronger rationale for the use of over-expression systems and the diverse cell lines chosen for different experiments (HEK293, U2OS, RCC cells).

Major comments:

- The statement in the abstract 'circadian rhythms are unexpectedly robust' in RCC should be clarified or further supported experimentally. This could include evidence that the circadian genes are indeed changing across circadian time in RCC cells or tumors.

We have added several new experiments to address circadian rhythmicity in RCC, which are described in response to a similar concern (point #3) that was raised by Reviewer 1. These new data are included in the revised manuscript Figures 1F-H and 8F and Supplemental Figures S2B, and S12- S14.

- It is known that overexpression of proteins can produce artificial interactions (Moriya, Mol Bio of the Cell, 2017). Fig. 1G would be strengthened through several changes: 1) through demonstration of the level of endogenous protein compared to overexpressed protein, 2) by demonstrating co-IP of canonical binding partners, e.g. BMAL1/CLOCK HIF2a/ARNT, 3) reciprocal pulldown of HIF2a and blotting for BMAL1.

We agree that overexpression of proteins can produce artificial interactions. Thank you for these suggestions. We have replaced the original Figure 1G with new Figure 3C in which we include co-IP of canonical binding partners and show that ARNT and ARNT2 pull down more endogenous HIF2 α than BMAL1 and BMAL2 do, while BMAL1 and BMAL2 pull down endogenous CLOCK but there is no detectable signal for CLOCK in co-IP samples with ARNT or ARNT2. This is consistent with all prior literature that supports the formation of BMAL1-HIF α heterodimers but not of ARNT-CLOCK heterodimers (See Hogenesch et al 1998; Vaughan et al 2020).

In addition, we generated a mutant BMAL1 (D144A) that greatly reduces interaction with HIF2 α without disrupting the BMAL1-CLOCK interaction (new Supplementary Figure 11A-D) and used it to demonstrate that interaction with HIF2 α plays an important role in the enhanced tumor growth of 786O xenograft tumors that overexpress BMAL1 (new Figure 8A-D). This finding is discussed in greater detail in response to Reviewer #4 (point g) below.

- While the CUT&RUN data presented in figure 4 are clear and interesting, more thorough analyses would greatly increase the impact of this experiment. For example, it is stated that the “Depletion of BMAL1 reduced chromatin association of both BMAL1 and HIF2 α at many sites that were occupied in control cells and reduced the number of significantly enriched loci detected in chromatin purified with BMAL1” (lines 165-166). However, the quantitative degree of change is not stated. Earlier in the paragraph, it is stated that “336 loci were identified as co-occupied by BMAL1 and HIF2 α , representing 18.5% or 27.9% of the sites associated with BMAL1 or HIF2 α , respectively” (line 154). This analysis is then expected for the data in which BMAL1 is knocked down. There should be an analogous percentage for those sites that were no longer occupied in the BMAL1 knockdown data on a global scale. This information would allow us to fully understand the degree of HIF2a and BMAL1 overlap and potential activity of the BMAL1/HIF2a heterodimer.

Thank you for these suggestions that spurred us to perform additional analyses that further strengthen our conclusions about the importance of BMAL1-HIF2 α overlap on chromatin. In 786O cells expressing *shBMAL1*, the BMAL1 cistrome was reduced as expected: Of the 1,813 BMAL1-associated peaks detected in *shControl*-expressing cells, 702 (38.7%) were detected in cells expressing *shBMAL1*. The intensity of the peaks that remained was reduced, suggesting that the remaining signal is coming from residual BMAL1 protein expression in *shBMAL1*-expressing cells. Of those 702 peaks, 98 were identified as co-occupied by BMAL1 and HIF2 α , representing 12.7% or 9.0% of the sites associated with BMAL1 or HIF2 α , respectively. So, there is much less overlap when BMAL1 expression is reduced. We have added this information to the text (lines 217-233).

- In Fig. 3E, ARNT knockdown caused large decreases in expression of hypoxia response genes, demonstrating ARNT has function in 786O cells. However, the HRE luciferase experiments were performed in U2OS cells, an osteosarcoma line, which likely has little relevance to ccRCC. In Fig. 6C, overexpression of HIF2a and ARNT in U2OS cells showed no increase in HRE reporter activity. Therefore, use of the inhibitor PT2399 on the HIF2a-ARNT interaction had no effect. It is unclear how to interpret this figure since HIF2a-ARNT do not drive expression of the HRE reporter. Furthermore, it is unclear if this data can be applied to ccRCC cells where previous figures demonstrated the importance of ARNT in hypoxia signaling in 786O cells.

We agree that the low activity of ARNT-HIF2 α in U2OS cells to activate the widely used HRE-Luc reporter that is derived from the *PGK1* locus makes interpretation of these assays difficult. We share the reviewer’s concern and appreciate their suggestion that the use of a non-kidney cell line may contribute to the lack of activity of ARNT-HIF2 α heterodimers. We now include new luciferase assay data performed in human embryonic kidney cells (HEK293T) in which we see similar levels of activation of the HRE-Luciferase reporter by ARNT-HIF2 α and by BMAL1-HIF2 α (new Figure 3D). This allowed us to perform a more meaningful examination of the impact of PT2399 on ARNT-HIF2 α vs. BMAL1-HIF2 α (new Figure 7D), which shows that BMAL1-HIF2 α

heterodimers are more sensitive to suppression by PT2399 than ARNT-HIF2 α heterodimers are.

In addition, we used quantitative RT-PCR to measure the expression of the BMAL1-CLOCK dependent transcript *NR1D1* and of HIF2 α target genes in response to PT2399 treatment of 786O cells expressing *shBMAL1* or a control shRNA (new figure 7E). In control 786O cells, PT2399 increases the expression of *NR1D1*, consistent with our model that PT2399 disrupts BMAL1-HIF2 α heterodimers and increases the availability of BMAL1 to associate with CLOCK. Depletion of BMAL1 by shRNA reduced *NR1D1* and abolished the effect of PT2399 on *NR1D1* expression. Conversely, the HIF2 α target genes *SLC2A14* and *ADM* exhibited decreased expression in response to PT2399. shRNA-mediated depletion of BMAL1 reduced the effect of PT2399 on *SLC2A14*, which we found to be primarily dependent on BMAL1 and had less impact on *ADM*, which we identified as primarily dependent on ARNT. Together, these data demonstrate that BMAL1 contributes to PT2399-driven transcriptional changes in ccRCC cells.

Minor comments:

- I suggest changing the title to be more precise and informative.

Thank you. We have updated the title as suggested.

- It is unclear how statistically significant BMAL1 expression is in Fig. 6A. Please provide fold change (resistant/sensitive) and p-values (or indication of statistical significance).

Thank you for your attention to detail. We elected to present these data in the volcano plot format to address their provenance from a genome-wide study and this obscures the significance of the expression change in *BMAL1* because there are other genes that have a larger fold change. We now show *BMAL1* expression in new Figure 7D and include the volcano plot as supplemental Fig. S10C to highlight that the change in *BMAL1* expression is statistically significant when accounting for genome-wide multiple hypothesis testing.

- There is little combinatorial analysis of CUT&RUN and RNA-seq. The only analysis is in Fig. 4E which shows pathway enrichment. Showing changes in expression of individual genes across multiple samples through a heatmap or qPCR validation of targets such as those shown in Fig. 4D would be helpful.

Thank you for this suggestion. We generated heatmaps depicting the changes in gene expression in response to *shARNT* and *shBMAL1* for genes associated with peaks for HIF2 α and/or BMAL1, which provides further support for the importance of BMAL1-HIF2 α heterodimers in supporting gene expression in ccRCC cells and the idea that ARNT and BMAL1 are alternative partners for HIF2 α to transactivate gene expression. Combinatorial analysis of the CUT&RUN and RNA-seq data are now included in the revised manuscript (new Figure 6) and described in the text in lines 235-267.

Reviewer #3 (Remarks to the Author): ECR, co-review with Reviewer #2

Thank you for your contribution to reviewing this manuscript and providing the constructive feedback that helped us to improve this study.

Reviewer #4 (Remarks to the Author): expert in HIF2 α in ccRCC, HIF2 α biology

Mello et al examine the functional interaction of BMAL1 with HIF2 α . Previous reports have established the interaction between hypoxia and circadian pathways and the interaction of HIF1 α with BMAL1. In this manuscript the authors provide evidence that HIF2 α -BMAL1 interact and they binding to overlapping sets of chromatin sites. Specifically, they show that BMAL1 recruits HIF2 α in specific loci and they outline consensus sequences of binding sites. Given the role of HIF2 α in RCC, they observe that BMAL1 is expressed high in RCC and provide evidence that HIF2 α -BMAL1 activate HRE promoters. They further investigated the role of BMAL1-HIF2 α dimers in growth of VHL deficient human RCC cells. Knock down of BMAL1 suppresses RCC cells growth in vitro and in vivo. Finally, they argue that BMAL1 expression confers sensitivity to PT2399 drug, based on in vitro observations only.

In Figure 3A, (KD western blot data), it appears that when BMAL1 is knockdown, ARNT protein expression increased. Reciprocally, KD of ARNT induced BMAL1 expression. Is this compensation mechanism?

Thank you for your attention to detail. We agree that the increased expression of endogenous ARNT upon depletion of BMAL1 and increase in endogenous BMAL1 in cells expressing shARNT could be a mechanism for compensation. We added a note to this effect in the text (lines 162-163). However, we have not studied this in enough detail to make a strong conclusion on this point.

The experiments in Figure 5 support the idea that BMAL1 contributes to the growth of VHL deficient cells in the colony formation assay and in vivo as xenografts, but they do not provide any evidence that this relates to the sensitivity to PT2399.

- a) Does shARNT or shBMAL1 shifts the sensitivity of the lines to PT2399?
- b) In the nude mice experiments are the BMAL1 KD tumors more or less sensitive to PT2399 in vivo?

Because ccRCC cells expressing *shARNT* or *shBMAL1* grow slowly and form very few colonies and very small (if any) xenograft tumors, it is difficult to assess their responsiveness to PT2399 either in 2D colony assays or *in vivo* xenografts. To address the reviewer's concern, we took two alternative approaches. First, we took advantage of

published crystal structures of BMAL1-CLOCK and ARNT-HIF2 α heterodimers to design and generate a mutant BMAL1 (D144A) that exhibits greatly reduced interaction with endogenous HIF2 α without reducing its interaction with endogenous CLOCK (Figure S11A-D). Overexpression of BMAL1 robustly increased the growth of xenograft tumors, while overexpression of BMAL1 D144A increased tumor growth to a much lesser extent (Fig. 8A and S11E). This indicates that BMAL1-HIF2 α heterodimers primarily drive the increase caused by BMAL1 overexpression. Furthermore, the xenograft tumors in which wildtype BMAL1 is overexpressed are highly sensitive to growth suppression by PT2399 but those that overexpress BMAL1 D144A were resistant to PT2399. Together, these findings demonstrate that BMAL1-HIF2 α heterodimers promote the growth of ccRCC xenografts and are sensitive to disruption by the HIF2 α antagonist PT2399 *in vivo*. The loss of sensitivity to PT2399 in xenograft tumors expressing mutant BMAL1 suggest that BMAL1 influences responsiveness to PT2399 but we can't make a strong conclusion about the mechanism underlying this observation and we have tempered our language on this point to avoid overstating our conclusions. These results are shown in Figure 8 and described in lines 309-328.

In parallel, we exploited the well-established circadian patterns of expression and transcriptional activity of BMAL1 to determine whether the response to PT2399 is correlated with the predicted contribution of BMAL1 to HIF2 α heterodimer activity *in vivo*. We treated mice harboring xenografts grown from 786O cells with PT2399 at two different phases of the circadian cycle and found that PT2399 is much more effective at suppressing xenograft tumor growth at ZT0 than it is when the mice are treated at ZT12. BMAL1 is more transcriptionally active at ZT0, so this is consistent with our model that BMAL1-HIF2 α heterodimers contribute to the effectiveness of PT2399 at suppressing the growth of ccRCC tumors. We cannot exclude the possibility that tumor-extrinsic physiological circadian rhythms in the host contribute to this divergent response. These results are shown in new Figure 8F and described in lines 330-361. We further discuss limitations of this experiment in lines 418-427.

c) In the colony formation assays it appears that ARNT is essential for 786O cell survival. However, from the Depmap data in Figure 1E, the BMAL1 shows significant dependency but not in ARNT. How do we reconcile this?

The DepMap data shown in Figure 1E are aggregated across 32 unique RCC cell lines, each of which shows variable dependency on ARNT and BMAL1. 786O cells are more dependent on ARNT than any of the other RCC cell lines tested in the DepMap database. While 786O cells are more dependent on ARNT than any of the cell lines is dependent on BMAL1, there are other RCC-derived cell lines that are not dependent on ARNT. For example, 769P cells have a positive dependency score for ARNT, meaning that they grow better than control cells when ARNT is deleted. While none of these cell lines is as dependent on BMAL1 as 786O cells are dependent on ARNT, their response to BMAL1 deletion is more consistently negative, resulting in a statistically significant dependency for BMAL1 but not for ARNT. We updated Figures 1E and 1F (now Figure 2B) to label the points that represent five of the 32 cell lines (including 786O) and amended the text (lines 114-125) to clarify this point.

The data presented in Figure 6 are supposed to support the idea that expression of BMAL1 confers sensitivity to PT2399. I have several concerns regarding this conclusion.

a) The compound is known to interrupt the HIF2a-ARNT interaction, nevertheless it is quite paradoxical that the HRE activity of co-expressed HIF2a-ARNT is NOT affected significantly by the compound (Fig 6C). Consistent with Figure 1H, BMAL1+HIF2a expression results in much higher HRE activity than HIF2a+ARNT, and subsequent significant reduction by PT2399. Does expression of BMAL1 affect HIF2a expression? How does the expression of ARNT compare with BMAL1 in the U2OS experiments? I am concerned that the different HRE activities induced by ARNT and BMAL1 are the result of different expression of exogenous bHLH-PAS proteins?

We agree that the low activity of ARNT-HIF2 α in U2OS cells on the widely used HRE-Luc reporter that is derived from the *PGK1* locus makes interpretation of these assays difficult. Based on the suggestion of Reviewer 2, we performed luciferase assays in human embryonic kidney cells (HEK293T) and now see similar levels of activation of the reporter by ARNT-HIF2 α and by BMAL1-HIF2 α (new Figure 3D). This allowed us to perform a more meaningful examination of the impact of PT2399 on ARNT-HIF2 α vs. BMAL1-HIF2 α which is now included as new Figure 7C and shows that BMAL1-HIF2 α heterodimers are more sensitive to suppression by PT2399 than ARNT-HIF2 α heterodimers are.

b) 786-O+BMAL1 number of colonies are reduced significantly by PT2399. How does this compare with 786-O+ARNT colonies? (please see combined response to points b and c)

c) BMAL1 knockdown suppresses the 786-O cell growth (Figure 5B and C). But why there is no effect of BMAL1 overexpression on colony formation?

Thank you for your thoughtful questions around these experiments that helped us to significantly improve our approach to understanding how altered expression of BMAL1 influences growth of ccRCC cells. We agree with the reviewers that two-dimensional colony formation assays with overexpressed proteins are not reliable for examining the contribution to PT2399 sensitivity. We observed significant variability in the impact of overexpression on colony formation without PT2399 treatment between experiments, and when the colony formation rates are altered by overexpression of ARNT or BMAL1, it is difficult to determine whether there is an additional impact on drug sensitivity.

For these reasons, we now include colony formation assays only when using genetic depletion of endogenous ARNT or BMAL1, which consistently and robustly reduce colony formation. To investigate how BMAL1 influences HIF2 α target gene expression, ccRCC xenograft tumor growth, and response to HIF2 α antagonist drugs, we now use two approaches. First we generated a mutant version of BMAL1 that greatly reduces its interaction with HIF2 α ; we compare the impact of overexpressing wildtype or mutant

BMAL1 on xenograft tumor growth and on response to PT2399 treatment *in vivo* (new Figure 8 and Supplemental Figure S11). In addition, we compared responses to treating mice with 786O xenograft tumors with PT2399 at two different times of day to ask how daily fluctuations in endogenous BMAL1 expression influence transcriptional and tumor growth suppression responses to PT2399 treatment *in vivo*. The difference in response was surprisingly large. These new data are included in new Figure 8 and Supplementary Figures S11 and S14.

d) Overall I am concerned that since BMAL1 may have a low expression in 786-O cells, these forced overexpression experiments may misrepresent the functional significance of BMAL1 with regards to HIF2a activity vis a vis to ARNT, especially if based on a single HRE reporter assay. For this reason, I think it is more appropriate for the authors to test by QRT-PCR specific HIF2a target genes, since they have a complete list of ARNTspecific, BMALspecific and common targets.

Thank you for this excellent suggestion. We used quantitative RT-PCR to measure the expression of the BMAL1-CLOCK dependent transcript *NR1D1* and of several HIF2 α target genes in response to PT2399 treatment of 786O cells expressing *shBMAL1* or a control shRNA (new figure 7E). In control 786O cells, PT2399 increases the expression of *NR1D1*, consistent with our model that PT2399 disrupts BMAL1-HIF2 α heterodimers and increases the availability of BMAL1 to associate with CLOCK. Depletion of BMAL1 by shRNA reduced *NR1D1* and abolished the effect of PT2399 on *NR1D1* expression. Conversely, the HIF2 α target genes *SLC2A14* and *ADM*, which we found to be dependent on BMAL1 or ARNT respectively, exhibited decreased expression in response to PT2399. shRNA-mediated depletion of BMAL1 tended to reduce *SLC2A14* as expected and had no effect on *ADM* in control cells. PT2399 did not reduce the expression of *SLC2A14* in 786O cells in which BMAL1 is depleted by shRNA. Together, these data demonstrate that BMAL1 contributes to PT2399-driven transcriptional changes in ccRCC cells.

e) I may misunderstand the concept here but how is it that high expression of BMAL1 provides PT2399 sensitivity but the gene appears minimally enriched in the sensitive PDX xenografts. How about ARNT expression level in sensitive or resistance samples?

We elected to present these data in the volcano plot format to address their provenance from a genome-wide study and this obscures the significance of the expression change in BMAL1 because there are other genes that have a larger fold change. We now show the expression of BMAL1 in sensitive and resistant xenografts in new Figure 7D and the expression of ARNT in the same samples in Supplementary Figure S10D. *BMAL1* expression is significantly higher in sensitive PDX samples but *ARNT* expression is not significantly different between the groups.

f) In Figure 6B the apparent “strong” effect of PT2399 on HIF2a-BMAL1 interaction is confounded by the fact that the FLAG-BMAL1 bait at 10microM is obviously less than the bait in the control lane. To quantify the effect and draw comparison between ARNT and BMAL1 we need triplicate experiments with densitometry measurements and

normalization to bait intensity.

Thank you for this excellent suggestion. We provide triplicate experiments with quantitation by densitometry measurements normalized to the bait intensity in new Figure 7A-B and Supplementary Figure S10B).

g) Lastly, to provide evidence that the proposed model is not artifactual or driven by HIF2a-independent program we need to test PT2399 sensitivity of a BMAL1 mutant that does not interact with HIF2a. Detailed published structure function analysis of ARNT should provide info for such a mutant.

Thank you for this excellent suggestion. We used the detailed structure information to design and generate a mutant BMAL1 (D144A) that exhibits greatly reduced interaction with endogenous HIF2 α without reducing its interaction with endogenous CLOCK (Figure S11A-D). Overexpression of BMAL1 robustly increased the growth of xenograft tumors, while overexpression of BMAL1 D144A increased tumor growth to a much lesser extent (Fig. 8A and S11E). This indicates that BMAL1-HIF2 α heterodimers primarily drive the increase caused by BMAL1 overexpression. Furthermore, the xenograft tumors in which wildtype BMAL1 is overexpressed are highly sensitive to growth suppression by PT2399 but those that overexpress BMAL1 D144A were resistant to growth suppression by PT2399. Together, these findings demonstrate that BMAL1-HIF2 α heterodimers promote the growth of ccRCC xenografts and are sensitive to disruption by the HIF2 α antagonist PT2399 *in vivo*. The loss of sensitivity to PT2399 in xenograft tumors expressing mutant BMAL1 suggest that BMAL1 influences responsiveness to PT2399 but we can't make a strong conclusion about the mechanism underlying this observation and we have tempered our language on this point to avoid overstating our conclusions. These results are shown in Figure 8 and described in lines 309-328.

h) What is the proposed mechanism by which BMAL1 promotes sensitivity to PT2399?

Our model is that BMAL1 promotes sensitivity to PT2399 because the BMAL1-HIF2 α heterodimer is more sensitive to disruption by PT2399 than the ARNT-HIF2 α heterodimer is. While it is well established that PT2399 can disrupt ARNT-HIF2 α , we show that it disrupts BMAL1-HIF2 α at a lower concentration and more robustly (Figures 7A-C).

Reviewer #1 (Remarks to the Author):

Overall, this revised manuscript includes substantial additional data compared to the previous version.

Thank you for your thoughtful comments that helped us to substantially improve this manuscript.

Reviewer #2 (Remarks to the Author)

The authors responded very well to the Reviewer's comments and the manuscript is much improved. With the new set of experiments added, we have a few minor comments:

- In the abstract, the statement, "Depletion of BMAL1 reprograms HIF2 α chromatin association", is unclear and the word "reprograms" should be more defined. The author's show that 393 new HIF2 α peaks emerge after BMAL1 knockdown (Fig. S8). However, the author's claim that these peaks do not "visually" constitute new HIF2 α peaks. Therefore, it is confusing if the point is that HIF2 α binds to distinct regions of chromatin in the absence of BMAL1 or not. We believe use of this strong language describing the relationship between HIF2 α and BMAL1 on chromatin should be softened. Another example of this language is, "BMAL1 plays an important role in directing HIF2 α to the genome." we do not think there is sufficient evidence to claim that BMAL1 "directs" HIF2 α to the genome since many new HIF2 α peaks emerge upon BMAL1 knockdown

Thank you for your close reading of our revised manuscript and for your further suggestions that will help to improve the communication of our findings. We agree that the word "reprograms" is not the best way to describe how depletion of BMAL1 alters HIF2 α chromatin association. We have changed the wording in the abstract to "Depletion of BMAL1 selectively reduces HIF2 α chromatin association" to more accurately reflect the impact of BMAL1 depletion on HIF2 α . We updated the wording later in the text to read "BMAL1 plays an important role in determining genomic localization of HIF2 α in ccRCC cells" to more accurately describe our findings.

- Fig. 1H shows antiphasic protein expression of BMAL1 and HIF2 α . Since this paper is on their function as a heterodimer, it seems paradoxical they would have antiphasic abundance. Can you address this point in the discussion or propose a mechanistic model?

We agree that the antiphasic protein expression of BMAL1 and HIF2 α has important implications for the role of BMAL1-HIF2 α heterodimers. This antiphasic expression means that the relative concentration of BMAL1 / HIF2 α will be much greater at the time of day when BMAL1 is high and HIF2 α is low, and thus BMAL1-HIF2 α heterodimers will be a much greater proportion of all HIF2 α heterodimers at that time of day. We believe this is an important contributing factor to the unexpectedly large effect of time of day

that we observed for responsiveness to PT2399 in vivo. We have addressed this point in the discussion with the following text added to the end of the second paragraph: “Notably, we found that rhythmic accumulation of BMAL1 and HIF2 α protein are approximately antiphase in 786O cells. This antiphasic expression means that the relative proportion of BMAL1-HIF2 α heterodimers compared to ARNT-HIF2 α heterodimers is much greater at the time of day when BMAL1 is high and HIF2 α is low. This likely contributes to the unexpectedly large effect of time of day observed for the suppression of xenograft tumor growth by PT2399 in vivo.”

- The rhythmic expression of HIF2a protein implies it is under circadian regulation, which is regulated either directly or indirectly by BMAL1-CLOCK. Therefore, it is necessary to show if BMAL1 knock down causes HIF2a protein abundance changes to interpret the genomic localization data in Fig 5 and S8. For example, if BMAL1 knockdown causes a decrease in HIF2a protein, then the reduction of HIF2a on chromatin in BMAL1 knockdown is likely due to non-specific abundance changes rather than BMAL1 “directing” HIF2a to certain genomic loci as is currently proposed.

Thank you for this important suggestion. We now added Western blots for HIF2a to Figure 4A and Supplementary Figure S6A, and we apologize for the oversight in not including these data previously. BMAL1 knockdown does not cause a decrease in HIF2a protein, if anything there is more HIF2a protein in 786O cells expressing *shBMAL1*. In A498 cells, depletion of ARNT reduces HIF2a protein accumulation and this may contribute to the decrease of HIF2a target gene expression upon loss of ARNT. We do not know the mechanism by which HIF2a protein is regulated by circadian clocks, but it does not appear to be through BMAL1-driven transcriptional regulation in this context. We suspect that it may involve CRY-mediated stimulation of HIF2a ubiquitination and protein turnover, but we have not tested this experimentally.

- Fig S11 E and F are missing in description in the figure legend.

Thank you for your close reading of our revised manuscript. We added the missing information to the legend for Figure S11.

- Although it's explained in the legend, it would be helpful to have an individual key for each graph of Fig. 8 since sometimes a filled dot means treated with PT2399 and sometimes it means vehicle.

Thank you for your close reading of our revised manuscript. We added an individual key to each graph in Fig. 8 to clarify which data represent tumors in mice treated with PT2399.

Reviewer #3 (Remarks to the Author)

I co-reviewed this manuscript with one of the reviewers who provided the listed reports. This is part of the Nature Communications initiative to facilitate training in peer review and to provide appropriate recognition for Early Career Researchers who co-review

manuscripts.

Reviewer #4 (Remarks to the Author)

I think the authors added crucial data to support their conclusions. The in vivo data and the uses of D144 mutant show that BMAL1 promotes tumor formation in vivo and that there is diurnal variation of the BMAL1 activity in RCC. They also toned down appropriately the statements in their discussion.

We appreciate the effort you made to provide feedback that helped us to improve this manuscript.

One very important detail that I think it is worth emphasizing regarding the mechanism is that the HIF2a-BMAL1 heterodimer is likely (based on their data) to execute an overlapping yet different program than the HIF2a-ARNT. In vitro some of the HIF target genes are NOT decreased by BMAL1, although their response to belzutifan is increased (Figure 7E NRD1 vs SCL or ADM, or SLC29A4 in Figure S9, if I read the data correctly).

We agree that the HIF2a-BMAL1 heterodimer likely executes an overlapping but different program related to that executed by the HIF2a-ARNT heterodimer. We have clarified and further emphasized this mechanistic model in the discussion section by updating lines 397-421.